

# Pre-seismic Thermal Anomalies from Satellite Observations: A Review

Zhong-Hu Jiao [1], Jing Zhao [2], and Xinjian Shan [1]

[1] State Key Laboratory of Earthquake Dynamics, Institute of Geology, China Earthquake Administration, Beijing 100029, China

[2] Harbin Institute of Technology Shenzhen Graduate School, Shenzhen 518000, China

*Correspondence to*: Zhong-Hu Jiao (jiaozhh@gmail.com)

**Abstract.** Detecting thermal anomalies prior to strong earthquakes is a key in understanding and forecasting earthquake activities because of its recognition of thermal radiation-related phenomena in seismic preparation phases. Data from satellite observations serve as a powerful tool in monitoring earthquake preparation areas at a global scale and in a nearly real-time manner. Over the past several decades, many new different data sources have been utilized in this field, and progressive anomaly detection approaches have been developed. This paper dedicatedly reviews the progress and development of pre-seismic thermal anomaly detection technology in this decade. First, precursor parameters, including parameters from the top of the atmosphere, in the atmosphere, and on the Earth's surface, are discussed. Second, different anomaly detection methods, which are used to extract thermal anomalous signals that probably indicate future seismic events, are presented. Finally, certain critical problems with the current research are highlighted, and new developing trends and perspectives for future work are discussed. The development of Earth observation satellites and anomaly detection algorithms can enrich available information sources, provide advanced tools for multilevel earthquake monitoring and improve short- and medium-term forecasting, which should play a large and growing role in pre-seismic thermal anomaly research.

## 1 Introduction

The earthquake is one of the most devastating and catastrophic natural disasters, which exposes human beings to tremendous risks (Geiß and Taubenböck, 2013). Therefore, strengthening the monitoring research of seismic activities using different technical means is necessary. Since the 1970s, remote sensing technology has been applied to seismic monitoring. In the 1980s, thermal infrared (TIR) satellite data were used for the first time to analyze thermal anomalies prior to earthquakes (Tronin, 1996 ). Earthquake monitoring based on passive or active remote sensing data is considered a promising research interest by the international seismology and remote sensing communities (Tronin, 2010), having the advantages of interdisciplinary research and being a new exploratory technology and tool.

Tectonic earthquakes are caused by the sudden dislocation of active faults due to surging tectonic stress (Freund, 2011). In addition to the considerable amount of strain energy released during the earthquake itself, the stress energy continuously accumulates during the preparation process of the earthquake. The activity of faults in earthquake-prone areas often results in



the growth of surface microfissures and gas ionization effects, following with changes in water content, underground gas, and earth electromagnetism around active faults. To some extent, these changes lead to pre-seismic thermal anomalies in seismogenic areas, such as regional warming and increased greenhouse gas concentration, which can be observed through satellite sensors.

Satellite remote sensing has its own advantages compared to traditional approaches of seismic hazard monitoring. Ground stations dedicated to earthquake monitoring are spatially relatively sparse; thus, their capability to monitor crustal movement and seismic information is limited. Meanwhile, global remote sensing data with high spatiotemporal resolution can be obtained through the advances in satellite technology. Satellite TIR data are a considerably reliable data source that exhibits the continuity, stability and accessibility. These data allow the study of the phenomena of large-scale and fast-changing seismic

thermal anomalies. Satellite TIR anomaly recognition can quickly capture thermal anomalies on the macroscopic scale, which contains thermal information of the Earth's interior caused by crustal movement. Moreover, satellite TIR data can reveal large-scale linear structures and short-term (i.e., from days to weeks) variations of thermal anomalies over tectonic plate borders and active faults (Filizzola et al., 2004; Tramutoli et al., 2005; Tronin et al., 2004). After more than 30 years of research, cases of thermal anomalies monitored by satellite remote sensing gathered from a large number of earthquakes have been repeatedly

tested and verified. Results show that thermal anomalies are present in seismogenic areas before earthquakes with the medium or strong magnitude.

Despite lots of promising research results about pre-seismic thermal anomalies, the thermal anomaly detection approach is still met with skepticism and controversies (Geller, 2011; Wyss 1997), because early warning signals from earthquake-prone regions are diverse, subtle, and often transient. Moreover, this approach is generally applied with some serious misjudgments

and misses earthquake prediction due to the unclear physical mechanism of pre-seismic thermal anomalies and perturbations from surface and atmospheric conditions and human activities (Tronin, 2010). Furthermore, the transfer process of surface thermal radiation is considerably complex, and numerous factors that affect the TIR radiation are highly varied. The recognition of pre-seismic thermal anomalies can be significantly hindered by topography, vegetation, soil moisture, surface type, and atmospheric conditions. Therefore, thermal anomalies caused by seismic activity sink into the background with various other

factors and are considerably weak relative to the surface thermal radiative variations caused by other natural or anthropogenic changes and thus can hardly be discerned. Seismic precursor identification has not yet formed an objective, stable and verifiable anomaly detection method; thus, it has not been completely accepted by the seismic community.

This paper attentively reviews the advances in pre-seismic thermal anomaly monitoring through satellite technology mainly over the last decade in terms of possible earthquake precursors and thermal anomaly detection approaches. Besides, it is noted

that reanalysis datasets can also be used in thermal anomaly analyses as mentioned in some studies below. These datasets are created via geophysical models and data assimilation scheme with spatial continuity and relatively low spatial resolution. The



satellite products and reanalysis data provide abundant data in this domain. Although inherent physical problems are always present, this work provides some assistant for attempts to detect pre-seismic thermal anomalies using satellite datasets.

The rest of this paper is organized as follows. Section 2 presents the geophysical variables used for pre-earthquake thermal anomaly detection in a number of earthquake cases studies. Section 3 describes several types of thermal anomaly detection methods, their basic properties, and recent applications. Section 4 discusses the issues around the pre-seismic thermal anomaly identification by satellite data products, which prove to be essential hindrances in earthquake prediction. Finally, Section 5 presents the expected development and perspectives for the next decades based on the review of current research status and trend, and analysis of existing problems in this domain.

## 2 Possible seismic precursors

All types of signals from the Earth-atmosphere system before strong earthquakes have been analyzed in a large amount of literature. The possible earthquake precursors used for anomaly detection involve various geophysical variables, such as seismic wave velocity, surface deformations, gravity, geoelectricity, geomagnetism, groundwater chemistry, temperature, moisture, atmospheric composition and ionospheric parameters (Cicerone et al., 2009; Tronin, 2006). Nevertheless, the pre-seismic anomaly analysis of precursors at a global scale and of high time-frequency has to rely heavily on satellite remote sensing datasets.

This paper focuses on thermal anomalies associated with seismic activities from above mentioned datasets. In the 1980s, after the discovery of thermal anomalies prior to earthquakes, seismic experts gradually realized the advantages of TIR remote sensing and conducted relevant studies. For more than 30 years, many geophysical variables, which can reflect thermal radiation information about the Earth's surface and/or atmosphere using satellite TIR observations and products (e.g., top-of-atmosphere (TOA) brightness temperature, outgoing longwave radiation (OLR), surface temperature, and latent heat flux) have been used to find a correlation between thermal anomalies and specifications of time, location and magnitude of imminent earthquakes. The following sections contain detailed explanations about these candidate physical quantities (Table 1)—from TOA, the atmosphere to the Earth's surface—and their applications in pre-seismic thermal anomaly analyses.

### 2.1 TOA Brightness temperature and OLR

The TOA brightness temperature is a measurement of the radiance of the TIR radiation at 3–5 μm or 8–14 μm from the Earth's surface, atmosphere, and clouds to satellite sensors. It is a direct and fundamental physical quantity expressed by the equivalent blackbody temperature of one channel (e.g., 11.77–12.27 μm band) in Kelvin. In other words, brightness temperature is not the normal temperature due to ignoring the emissivity effect. Many satellite sensors obtain TIR measurements, such as the Moderate Resolution Imaging Spectroradiometer (MODIS), Visible Infrared Imaging Radiometer Suite and Advanced Very





High Resolution Radiometer (AVHRR) aboard polar-orbiting satellites, and Spinning Enhanced Visible and Infrared Imager and Advanced Himawari Imager aboard geostationary satellites.

TOA brightness temperature data are one of the mostly used remote sensing data in thermal anomaly detection because of their easy accessibility and data processing. Several studies indicate that significant brightness temperature anomalies existed in seismically active areas about one month to eight days before earthquakes (Lisi et al., 2010; Lisi et al., 2015; Pergola et al., 2010). The regions with strong tectonic activities also showed intensive TIR anomalies during the preparatory phases of low-magnitude earthquakes (Lisi et al., 2010). After a 10-year retrospective correlation analysis in Greece, TOA brightness temperature anomalies are considered to satisfy the qualifications involved in a multi-parametric system for the time-dependent assessment of seismic hazards (Eleftheriou et al., 2016). As shown in Fig. 1, thermal anomalies mainly occurred before one earthquake with magnitude greater than 5, and for seismic events with magnitude less than 5 the indicator of thermal anomalies became less significant.

Another frequently used precursor at TOA is OLR, which is a critical component of the Earth's energy budget. OLR is the total longwave radiative flux (W/m²) in the TIR wavelength from 4 μm to 100 μm, which is emitted from the Earth and its atmosphere going to space. More specifically, it is the integrated result of the reflection, absorption, and emission of longwave radiation by the Earth's surface (land and sea), atmosphere, aerosols and clouds, and finally leaves the Earth system. OLR represents the radiation characteristics of the cloud top when the instantaneous field of view is covered by clouds and indicates the longwave radiation emitted by the surface and atmosphere under cloudless conditions.

OLR cannot be directly measured but can be derived from measurements of sensors (e.g., AVHRR, Atmospheric Infrared Sounder (AIRS), and Clouds and the Earth's Radiant Energy System (CERES)). Among them, the National Oceanic and Atmospheric Administration (NOAA) OLR is the commonly used data product in various studies and is retrieved from the AVHRR TIR bands (10.5–12.5 μm) on a 2.5°× 2.5° latitude–longitude grid with daily and monthly mean values from 1981 to the present (Gruber and Krueger, 1984). The latest AIRS OLR product, which is retrieved from hyperspectral TIR data, is AIRS Version 6 Level 3 monthly mean product on a 1°× 1° latitude–longitude grid beginning from September 2002 (Aumann et al., 2003). Finally, the CERES OLR dataset is the monthly means from 2000 in the CERES energy balanced and filled edition 4.0 datasets on a 1°× 1° latitude–longitude grid (Rose et al., 2013).

Several studies suggest that OLR can be effectively used as a short-term or impending earthquake precursor. OLR appeared as a steadily increasing trend around the epicentre and thus can be used as an indicator of seismic activity (Ouzounov et al., 2007). Fig. 2 presents the time series of daily OLR anomaly before Mw 9.0 Sumatra Andaman Islands, Northern Sumatra earthquake (3.09°N, 94.26°E) occurred on December 26, 2004. After several small peaks beyond the mean field of OLR, an intense rise showed up shortly following this mega earthquake. Anomalous OLR occurred two months to several days prior to the main shock with at least several thousands of square kilometers. OLR anomaly has a direct significance for earthquakes





with magnitude above 5.5 (Eleftheriou et al., 2016). Anomalous OLR variations can reach 30–45 W/m$^2$ around epicentral areas 7–8 days prior to the major earthquake (Rawat et al., 2011). Cumulative tectonic stress in the earthquake preparation regions may result in intensive thermal emissions associated with the release of a large amount of energy. Increase in surface temperature, air temperature, water vapor and other greenhouse gases result in the increase in OLR and TOA brightness temperature, especially around the epicenter under clear sky condition (Tramutoli et al., 2005).

TOA brightness temperature and OLR represent longwave radiation variations of the entire surface-atmosphere system, and are significantly affected by atmospheric conditions. The height and coverage of clouds have a dominating influence on the total intensity of longwave radiative fluxes received at the TOA (Turner et al., 2015). Therefore, eliminating cloud effects is necessary in applications, which can be difficult when working with low-resolution data. Meanwhile, thermal anomalies derived from OLR may be the result of other atmospheric effects and not of increased surface temperature (Tronin, 2010). In the cases under clear sky condition, water vapor is another significant factor that has strong absorption and re-emission capabilities on longwave radiation and rapidly changes in the spatiotemporal pattern. These situations are particularly unfavorable for detecting weak pre-seismic thermal anomalies; thus, surface land temperature (LST) and similar data products that are not heavily affected by atmospheric disturbances are preferred (Filizzola et al., 2004; Lisi et al., 2015).

### 2.2 Atmospheric water vapor and temperature

Atmospheric water vapor and temperature are two important physical quantities that describe the status of the atmosphere, especially in the atmospheric boundary layer. They have strongly dependencies and thus can present synchronous responses to the impending earthquakes.

Over 99% of atmospheric moisture is in the form of water vapor, and it is abundant and highly varies in abundance in the atmosphere. Water vapor is the primary greenhouse gas in the Earth's climate system and influences long-term climate changes. The water vapor associated with atmospheric energy drives the development of weather systems and the transfer of heat from the tropics to the poles. Atmospheric water vapor content can be referred to as the total column water vapor contained in a vertical column of atmosphere between the Earth's surface and the TOA, which can be retrieved from near-infrared, TIR or passive microwave data (Chesters et al., 1983; Ferraro et al., 1996; Gao and Kaufman, 2003). For example, the MODIS total precipitable water product (MOD05_L2 and MYD05_L2) has two types of data, one using near-infrared algorithm at the 1-km spatial resolution during the day, and the other using infrared algorithm at 5-km resolution during day and night. In general, the accuracy of water vapor products ranges from 5% to 10% (Gao and Kaufman, 2003).

Anomalies of atmospheric water vapor content have been observed prior to several strong earthquakes (Cervone et al., 2004; Dey and Singh, 2003). The 2001 Gujarat earthquake triggered anomalous atmospheric moisture increase of about 15 mm, which were detected using passive microwave data of the Special Sensor Microwave Imager (Dey et al., 2004). The anomalous increase of water vapor level occurred around the epicenter in land regions before the earthquake and in ocean regions after



the earthquake. Meanwhile, AIRS data led to the discovery of anomalous changes in relative humidity at different pressure levels reaching 500 hPa with even larger variations at the lower levels (Singh et al., 2010a). This phenomenon may be related to the increase of surface latent heat flux (SLHF) (Dey et al., 2004), which is discussed in Section 2.5.

Aside from atmospheric moisture, air temperature (or atmospheric temperature) is the most commonly measured weather parameter. It refers to the screen-level air temperature in meteorological terms, which is normally measured at the attitude of 2 m above the ground surface. Air temperature is a more complex parameter than LST, which is affected by LST, winds, topography, etc. LST can be estimated based on satellite TIR measurements. However, air temperature cannot be directly derived from LST, and no operational remote sensing data product of air temperature is available at present. The atmospheric profiles provide the air temperature and water vapor amounts at different altitudinal layers in the vertical atmosphere, but this air temperature is different from the screen-level temperature. The atmospheric profiles can be retrieved from TIR multispectral sensors (e.g., MODIS) or infrared sounders (e.g., AIRS) (Aumann et al., 2003; Seemann et al., 2003).

Significant near-surface air temperature anomalies related to earthquakes have been reported in different studies (Panda et al., 2007; Pulinets and Dunajecka, 2007). An approximate increase of air temperature at 2–4 K in a north-south pattern along the Nosratabad fault zone during 18–20 December was observed prior to the Ms 6.0 Kerman earthquake in Iran at December 20, 2010 (Fig. 3) (Alvan et al., 2013). Anomalous changes of air temperature at different pressure levels from AIRS data were found up to 500 hPa with the variations becoming larger at the lower levels (Singh et al., 2010a). An increase in air temperature along with low air humidity has been observed prior to earthquakes (Pulinets et al., 2006; Singh et al., 2010a). The anomalies in air temperature and relative humidity prior to strong earthquakes indicated by historical meteorological data are primarily related to the air ionization caused by increased radon emanations from faults in earthquake preparation regions (Ouzounov et al., 2007; Pulinets and Dunajecka, 2007). Anomalous areas are considerably larger than seismic sources, which indicates the difficulty of identifying the epicenter before a main shock.

### 2.3 Atmospheric trace gases

Atmospheric trace gases make up less than 1% of the Earth's atmosphere by volume. These gases include $CO_2$, $CH_4$, $O_3$, $SO_2$, and $NOx$, which are produced from natural geothermal sources, volcanic activity, ecosystem and anthropogenic sources. Trace gas concentrations are crucial in investigating atmospheric processes. Atmospheric trace gas data, such as $CH_4$, $CO_2$ and CO, can be retrieved from satellite measurements (e.g., AIRS hyperspectral TIR data) (Chahine et al., 2006; Clarisse et al., 2011).

Seismic activities can change atmospheric composition, especially the increment of greenhouse gases prior to strong earthquakes. A 6% increase in the columnar ozone prior to the 2015 Nepal earthquake was observed (Ganguly, 2016). An increased CO emission at near-surface pressure levels associated with LST over epicenter areas was detected by the Measurements of Pollution in the Troposphere onboard Terra satellite approximately one week before the earthquake (Singh et al., 2010b). Anomalous increases in total column CO and $O_3$ occurred over the epicenter areas one to six months prior to





several strong earthquakes (Cui et al., 2013). Fig. 4 indicates that anomalous CO appeared along the San Andreas Fault, while
the anomalies of $O_3$ spread over wider areas around the epicenter before and after Ms 7.1 Baja California earthquake on 5
April 5, 2010. However, these phenomena are unstable with only parts of studied earthquakes exhibiting anomalous variations
and with persistent time being irregular.

Increased pressure between rocks due to seismic activity can result in greenhouse gas emissions and local greenhouse effect
(Choudhury et al., 2006; Singh et al., 2010a; Zoback and Gorelick, 2012). The experimental results show that gases, such as
$CH_4$ and $CO_2$, can obtain energy from the transient electric field and cause a temperature increase. The thermal anomaly before
medium to strong earthquakes was preliminarily believed to exhibit 1) a sudden release of gas and 2) a static electricity of field
mutation. Earthquakes may lead to the release of underground carbonaceous gas, which provides a scientific basis for the
mechanism of pre-seismic thermal anomalies. In addition, $CO_2$-rich fluids in the deep crust are considered as the possible
important carbonaceous gas source in the lithosphere–coversphere–atmosphere coupling (LCAC) process (Chiodini et al.,
2011). The anomalies in trace gas concentration can be attributed to gas emissions in the lithosphere and photochemical
reactions (Cui et al., 2013). As a result, LST and air temperature may be increased, resulting in changes in SLHF and
atmospheric water vapor condensation, and propagation to the brightness temperature observed by satellite sensors.

**2.4 Aerosols**

Aerosols are the fine solid and liquid particles suspended in the atmosphere, which are produced from various sources, such
as desert dust, wildfire smoke, sea salt particles, volcanic ash, and urban haze. Aerosols have different optical and
microphysical properties depending on their type and size distribution. They have an important influence on cloud formation
and are often expressed by aerosol optical thickness (AOD). AOD is a dimensionless physical quantity used to describe
quantitatively the extinction degree of the direct solar radiation reaching the ground by the absorption and scattering of aerosols.
AOD is defined as the integrated extinction coefficient within a vertical column of unit cross section along the entire
atmospheric path, where the planetary boundary layer accounts for most of them. For now, most of multispectral radiometers
in the visible, near-infrared and shortwave-infrared wavelength regions can be used to retrieve AOD. For instance, the MODIS
daily Level 2 aerosol product (MOD04), which is produced by the "Dark Target" and "Deep Blue" algorithms at a spatial
resolution of 10 km at nadir in the Collection 6 (Levy et al., 2013), includes many aerosol parameters, such as AOD, aerosol
type, angstrom exponent, asymmetry factor, and single scattering albedo.

Aerosols are used as indicator in many per-seismic anomaly studies. AOD was observed an increase of 40% prior to the
2015 Nepal earthquake (Ganguly, 2016). A clear increase in AOD around the epicenter 2 days prior to the 2010 Chile
earthquake was found in Fig. 5 using three anomaly detection methods in order to reduce uncertainty of detecting anomaly
date (Akhoondzadeh, 2015). An anomalous increase of ground-measured AOD is quasi-synchronous with several abnormal
hydrothermal parameters (Wu et al., 2016). Over the sea surface, significant changes in aerosol parameters are considered to



be related to the 2001 Gujarat earthquake (Okada et al., 2004). Although aerosols don't directly relate with the TIR radiation, the charged aerosol particles play a role in the water vapor condensation and cloud formation over earthquake fracture areas. A high aerosol concentration causes stronger electric field intensities and increased infrared emissions (Liperovsky et al., 2005). Therefore, indirect effects of aerosols on thermal anomalies cannot be ignored.

## 2.5 SLHF

SLHF ($W/m^2$) is the flux of the heat absorbed or released by the phase transition (i.e., condensation, evaporation, and melting) of water from the Earth's surface to the atmosphere. SLHF is an important component of Earth's surface energy budget and is mainly affected by the atmospheric relative humidity, wind speed, surface temperature, and season. Traditionally, SLHF is measured through the Bowen ratio technique or by the eddy covariance, and can also be obtained from satellite products. MODIS global evapotranspiration product (MOD16) has a spatial resolution of 1 km at eight-day, monthly, and annual

intervals (Mu et al., 2011).

Several studies have reported the abnormal increase of SLHF prior to earthquakes. The SLHF increased by approximately 50 $W/m^2$ peaking at 115 $W/m^2$ over 3–23 days before earthquakes in the Middle East (Mansouri Daneshvar et al., 2014). SLHF can also be a possible precursor to coastal earthquakes because anomalous SLHF tends to be more intensive over sea surface than over land surface (Cervone et al., 2004; Cervone et al., 2006). The analysis of coastal earthquakes shows that the SLHF

increased abnormally a few days before earthquakes, which is the result of the increase in surface temperature in seismically active areas (Pulinets et al., 2006). SLHF anomaly is significantly weaker on land surface than on sea surface due to obvious differences in air humidity, wind speed, and surface temperature between sea and land surfaces. However, Qin et al. (2014a) found that anomalous SLHF, which is indicated by the dashed circle in Fig. 6., occurred approximately 10 days before the inland Pu'er earthquake (23.8092°N,101.15°E) on 2 June, 2007. Conversely, the study of Zhang et al. (2013) indicates that no

significant anomalous variation of SLHF exists prior to ten studied marine or coastal earthquakes, and that data accuracy and parameter settings have an important effect on quantification of the correlation between SLHF anomaly and earthquake's occurrence.

SLHF anomaly prior to earthquakes is related to the underground fluid movement and the interaction among the underground, surface and atmosphere (Alvan et al., 2013). Mansouri Daneshvar, *et al.* believed that water vapor responses to seismic

preparation phases follow a seismic-triggered chain, which undergoes air ionization, SLHF enhancement, water vapor condensation, and increased rainfall (Mansouri Daneshvar et al., 2014). Several studies have proven that SLHF anomalies before earthquakes are linked with soil moisture (Cervone et al., 2004; Dey and Singh, 2003). Meanwhile, air ionization in the lithosphere–atmosphere–ionosphere coupling (LAIC) model may be responsible for soil moisture and SLHF anomalies due to the increase of water condensation on newly formed ions (Pulinets and Ouzounov, 2011).





### 2.6 Surface temperature

LST, also known as surface skin temperature, represents a remarkably thin surface layer of medium temperature state, which is affected by solar radiation, surface albedo, vegetation cover, soil moisture, etc. LST affects the energy exchange between the surface and boundary layers and determines the air temperature near the surface. Compared to TOA brightness temperature and OLR, LST can indicate the surface warming process better. MODIS LST data are currently the most extensively used products in the 1-km spatial resolution with an inversion accuracy of approximately 1 K (Wan, 2014).

LST is suitable for the analysis of inter-monthly variations, which is beneficial to short-term earthquake monitoring. A 40-day time period is sufficient for anomaly monitoring (Bhardwaj et al., 2017). The LST prior to earthquakes usually increased by at least 3–5 K (Ouzounov et al., 2006; Ouzounov and Freund, 2004; Tronin, 2006). A 1–10 K rise of LST two weeks before the main shock of 2011 Tohoku-Oki earthquake was detected around the epicenter (Zoran, 2012). Moreover, the study by (Bhardwaj et al., 2017) shows that snow cover levels prior to the 2015 Nepal earthquake noticeably declined and that snow surface temperature is a highly sensitive precursor that presented 3–14 K temperature anomaly in 3–17 days prior to this earthquake. Most studies use nighttime LST data for analysis due to the absence of solar heating, leading to measurements that are predominantly gathered from surface and underground thermal sources. However, daytime LST data can also show significant thermal anomalies (Blackett et al., 2011; Panda et al., 2007; Saraf et al., 2008).

The thermal anomalies in epicentral areas before strong earthquakes are related to the increase in air temperature, LST and soil temperature at shallow layers. LST anomaly indicated by MODIS data was consistent with anomalous air temperature variations observed at the meteorological station close to the epicenter (Akhoondzadeh, 2013). The anomalous temperature variations are related to the seismic deformation and follow the fault evolution under seismic stress changes (Ma et al., 2008). Thus, LST is more useful in detecting thermal anomalies than TOA brightness temperature or OLR because it is less affected by the atmosphere, which allows it to produce a higher signal-to-noise ratio (SNR) (Aliano et al., 2008; Lisi et al., 2015). Fig. 7 shows the spatial and temporal evolution of the sequence of LST anomalies by different colored dashed circles (Lisi et al., 2015). The green dashed circle indicates the possible local warming effect caused by clouds. The LST anomalies which followed the main tectonic lineaments (green lines) in Central Italy (black circle), the Balkan region (blue circle) and the Padania plain (purple circle) were possibly associated with the medium-strong earthquakes. Meanwhile, LST, as a thermodynamic parameter of dynamic equilibrium, derived by remote sensing data contains a degree of uncertainty due to the influence of surface types, surface energy balance process and atmospheric turbulence.

Several cases of marine earthquake monitoring using remote sensing TIR data have been successful. Sea surface temperature (SST) anomaly can be related to seismicity (Ouzounov and Freund, 2004), but SST is strongly influenced by weather conditions and currents. Seawater has a large thermal inertia that allows its temperature to change more slowly, therefore the mechanism of LST anomalies is not applicable to SST anomalies. The study of the 2000 Blanco earthquake sequence shows





that ocean hydrothermal system can be altered by large earthquakes and significant decline of temperature was observed following the main shock (Dziak et al., 2003). Ouzounov, *et al.* analyzed the decrease of SST before the Ms6.8 2003 Boumerdes earthquake and implied that this phenomenon was attributed to the upwelling of cold water from the deep ocean caused by crustal deformation (Ouzounov et al., 2006).

## 3 Anomaly detection methods

Thermal anomaly can generally be defined as the departure of a current observation value from the long-term average reference (Ouzounov et al., 2007); it is used to relate seismicity with temporal and spatial variations of thermal radiation signals from satellite observations. Different data processing methods have been proposed in this domain over the past years. These methods aim to detect pre-seismic thermal anomalies, attempt to point out possible future strong earthquake in terms of time, spatial, and magnitude windows within stated limits, and estimate the probability of this earthquake's occurrence. Fig. 8 presents the methods that are classified and discussed below.

### 3.1 Visual interpretation

Visual interpretation method is a type of manual analysis method that interprets the relationship between an earthquake and the relevant parameters through a time series of visible and near-infrared or TIR data before and after the target earthquake. The observed quantity, such as surface temperature, is measured in considerably greater amounts prior to the earthquake around epicenter areas compared to other time period and neighboring regions. The method is simple and effective; thus, it is always used in interpreting preliminary pre-seismic anomalies. Several studies have utilized this method, such as in the aerosol anomaly (Qin et al., 2014b), SHLF anomaly (Dey and Singh, 2003), anomalies of multiple parameters (Pulinets et al., 2006; Singh et al., 2010a), and thermal anomalies in AVHRR and MODIS brightness temperature (Saraf et al., 2012).

Visual interpretation is mainly based on experience interpretation, which results in strong subjectivity and does not allow current computer technology to deal with massive amounts of satellite data. From Fig. 9, we can distinguish the LST anomalies around the epicenter, but this difference is subtle causing the randomicity during thermal anomaly recognition. In addition, visually identifying abnormal information can be difficult and calls for further processing, such as the use of pseudo-color synthesis, difference method, and fault profile line method. At the same time, increasingly sophisticated anomaly detection algorithms have been applied to extract signals about pre-seismic thermal anomalies with increases in data source and amount and the quantitative requirement of further research.

### 3.2 Image difference methods

The simple image difference method involves identifying the differences in the same region over a series of images captured at different times for qualitative analysis. If the difference exceeds the pre-defined threshold, then the observed precursor value





could be regarded as a thermal anomaly that may be associated with an impending earthquake. This method eliminates the influence of background signals and highlights the spatial distribution of thermal anomalies to a certain extent. For example, TIR images captured before and after earthquakes were used to calculate difference values that highlight thermal anomalies (Ma et al., 2008; Tronin et al., 2002). The method that uses brightness temperature difference has a certain ability to identify cloud pixels, but cannot eliminate background influence. Therefore, temperature anomaly is introduced through normalizing

the area-averaged LST by subtracting and dividing by multiyear mean LST (Zoran, 2012).

The general anomaly method is to find the difference between current data during an earthquake and the long-term average value (also called a reference value) in order to improve result robustness and eliminate potential bias (Mahmood et al., 2017; Qin et al., 2014a; Wu et al., 2016). This method can be expressed as

$$GA(x, y) = v(x, y, t) - \mu(x, y), \tag{1}$$

$$\mu(x, y) = \frac{1}{N} \sum_{t=1}^{N} v(x, y, t), \tag{2}$$

where $v(x, y, t)$ is the value of a pixel at a spatial location $(x, y)$ and time $t$ of satellite data required, which can be TOA brightness temperature, OLR ,or LST; and $\mu(x, y)$ is the average value at the same position in the same or similar time slot of years using multiyear data. In other words, the anomaly is the difference between what is happening and what is expected over a long-term average trend. General anomaly method is an improvement of the image difference method (Qin et al., 2012).

Given complex influences and possible uncertainties with respect to seasons, local weather, topography, and surface inhomogeneity, the pre-seismic thermal anomaly is a weak signal in the strong background level. Thus, understanding variations of the background field and the variation trend of each influencing factor over time are the key to thermal anomaly detection. The establishment of regional reference field based on multiyear data is a basic procedure to identify pre-seismic thermal anomaly. Because reference field can express the trend of physical quantity in the spatiotemporal pattern and relieve

random variability because of local meteorological factors. General anomaly method is used to eliminate this tendency change of current observation, thereby highlighting thermal anomalies. The $k$ times standard deviation ($\delta$) is generally the reference level relative to the long-term normal field. Beyond $\pm k\delta$ from the mean value of the reference field is regard as the anomaly. A positive anomaly means that the value is higher than the normal trend (warming effects), whereas a negative anomaly indicates a lower value than the normal value (cooling effects).

The thermal anomaly is a weak signal under strong interference, thus isolating it by simple comparison or threshold is difficult. This may also be an important reason that not all earthquakes exhibit significant pre-seismic thermal anomalies. The eddy field method is used to calculate and add the difference data between adjacent lattice points of the data to obtain the spatial distribution of the vorticity value (Ouzounov et al., 2007). The formula is as follows:

$$EF(x, y) = 4 \cdot v(x, y) - v(x - 1, y) - v(x + 1, y) - v(x, y - 1) - v(x, y + 1), \tag{3}$$





where $EF(i, j)$ is the eddy value at spatial location $(x, y)$, and $v$ is the observation value of a pixel. This method can highlight

anomalous changes in the neighborhood and is used for earthquake-related anomaly detection by NOAA OLR data (Ouzounov

et al., 2007; Xiong et al., 2010).

Furthermore, normalization by Z-score (ZS) is as an index of thermal anomaly, which hereafter is referred to as the ZS

method (Ouzounov et al., 2011). The ZS index is defined as follows:

$$ZS(x, y, t) = \frac{v(x,y,t) - \mu(x,y)}{\delta(x,y)},\tag{4}$$

$$\delta(x, y) = \sqrt{\frac{1}{N-1} \sum_{t=1}^{N} \big( v(x, y, t) - \mu(x, y) \big)^2},\tag{5}$$

where $\delta(x, y)$ is the standard deviation of this reference field. This method is similar to the robust satellite technique (RST)

method, which introduces the concept of SNR. The difference between current and mean values of a reference field represents

the signal part, and the standard deviation of a reference field is the noise part. A higher SNR indicates a greater possibility of

thermal anomalies prior to an earthquake (Tramutoli et al., 2001).

### 3.3 RST method

At present, RST, which is extensively used in thermal anomaly detection, is a multi-temporal statistical method for analyzing

long-term satellite records with similar observation conditions (e.g., same month, same time of day, and same sensor data)

(Genzano et al., 2015; Lisi et al., 2010; Pergola et al., 2010; Tramutoli et al., 2005). RST combines the neighborhood difference

in the eddy field method and the normalization of the reference field in the ZS method, and it clearly defines the mathematical

expression of the thermal anomaly in statistics. This method considers the statistical characteristics of the historical data of

seismically active areas to discriminate the possible pre-seismic abnormal behavior of current thermal signals. It is also

considered applicable to different atmosphere and surface conditions and different satellite observation geometries, and can

reduce the probability of occurrence of false thermal anomalies (Filizzola et al., 2004; Pergola et al., 2010).

The robust estimator of TIR anomalies (RETIRA) index is used to express the intensity of thermal anomalies in RST method,

as shown as follows:

$$\otimes_{\Delta v}(x, y, t) = \frac{\Delta v(x,y,t) - \mu_{\Delta v}(x,y)}{\delta_{\Delta v}(x,y)},\tag{6}$$

$$\Delta v(x, y, t) = v(x, y, t) - \bar{v}(t),\tag{7}$$

where $\bar{v}(t)$ is the spatially average value in the homogeneous neighborhood; such that $\Delta v(x, y, t)$ represents the difference

between the value of a pixel in $(x, y)$ and ambient homogeneous pixels; $\mu_{\Delta v}(x, y)$ and $\delta_{\Delta v}(x, y)$ are respectively the mean and

standard deviation in $(x, y)$ of the reference field calculated from $\Delta v(x, y, t)$ in the same or similar time slot from many years,

which are referred to formulas (2) and (5). The RETIRA index is further used to indicate the relationship with future seismic

activity. Generally, a value greater than 3 to 4 is used as the threshold for the presence of thermal anomalies that are associated

with possible earthquake occurrences (Eleftheriou et al., 2016; Lisi et al., 2010; Lisi et al., 2015).





If continuous and large-scale thermal anomalies appear from a time series of images within study areas, then they can be considered pre-seismic thermal anomalies (Aliano et al., 2008). Therefore, an index of significant sequences of thermal anomalies (SSTAs), which measures the persistence and continuity of thermal anomalies in temporal and spatial patterns, is defined to remove problematic values as far as possible (Genzano et al., 2015; Lisi et al., 2015). SSTAs standards include the following: 1) relative strength: RETIRA index is greater than or equal to 3; 2) spatial persistence: the area of thermal anomalous

pixels in the region of $1°×1°$ is larger than 150 km$^2$; 3) temporal continuity: the phenomenon of thermal anomalies occurs at least once within 7 days before or after current time.

The purpose of SSTAs is similar to that of a deviation-time-space-thermal (DTS-T) anomaly recognition method proposed by Wu et al. (2012), which is applied to analyses of several large earthquakes. The DTS-T method defines a normalized quantitative index to express the stability of thermal anomalies considering the significance of numerical differences,

synchronization of time, and adjacency of geographical location. Both indices are intended to improve the stability of detected thermal anomalies that are related with impending earthquakes as much as possible. In addition, geostationary satellite data are more advantageous than polar-orbiting satellite data in thermal anomaly detection (Aliano et al., 2008; Lisi et al., 2015). Because it has a higher temporal resolution and a fixed observation geometry, and thereby its data have a higher SNR in detecting seismically related thermal anomalies when using RST-based method. The principle of this algorithm is concise;

however the impact of short-term meteorological warming is difficult to eliminate. Therefore, this method should be combined with other data to identify the interpretation of anomaly detection results.

**3.4 Methods based on wavelet transform**

Wavelet analysis is one of the most popular seismic anomaly detection methods in recent years. Wavelet decomposition, wavelet packet, and power spectrum are used in determining the transformation and decomposition of TIR signals to extract

useful feature signals for identifying pre-seismic thermal anomalies. Wavelet transform has the feature of multiresolution analysis, which can characterize the local characteristics of the signal in time-frequency domain.

The influence factors of surface thermal radiation are complex and varied; therefore, influences of various field sources are difficult to separate. Furthermore, non-tectonic factors, such as landforms, land covers, and meteorology, always have greater effects on thermal radiation than tectonic movements or thermal radiation induced by earthquakes. The weak signal under

strong interference is difficult to separate, resulting in unstable results of anomaly detection methods. Continuous TIR radiation data can be considered to include the long-period components (e.g., the basic radiation field of the Earth, the inter-annual or seasonal radiation field, and terrain induced radiation field), the short-period components (e.g., diurnal radiation field, cloud and cold air flow in several hours to days, and thermal radiation components caused by tectonic movements or earthquakes), and the last residual factors (e.g., high-frequency noise components). Wavelet transform can decompose the time series signal





into signals of different frequency bands that correspond to above different components; thus, studying the energy variation in each frequency band is convenient.

The relative power spectrum (RPS) method performs wavelet transform on time series data to realize bandpass filtering, remove long- and short-period components, and retain certain intermediate frequency band information that may be connected with seismic-related signal changes and is expressed in the form of power spectrum. RPS method was used to analyze thermal

anomalies of the 2008 Wenchuan earthquake using TOA brightness temperature data (Zhang et al., 2010). Fig. 10 presents the obvious brightness temperature anomalies from April 25 to June 19, 2008 over the Longmenshan fault zone, and demonstrates the good results of the RPS method. The wavelet transform is initially used to process the multiyear data after cloud filtering and filling in of missing data. For each pixel, the basic temperature field, annual variation temperature field, daily variation temperature field and other factors can be separated, and remaining information may include earthquake-related

information. The RPS is then calculated by Fourier transform with a certain time window and sliding window lengths. On the basis of TOA brightness temperature, RPS approach is used to analyze pre-seismic thermal anomalies, but non-seismic thermal anomalies always appeared (Xie et al., 2013).

The wavelet transform method is used to eliminate low-frequency annual and seasonal components and high-frequency random components, such as local meteorological conditions and human activities. The local maxima are then extracted from

time series of MODIS LST data by a predefined threshold, which is sensitive to peak changes (Saradjian and Akhoondzadeh, 2011). Another wavelet maxima method proposed by (Cervone et al., 2004) uses one-dimensional wavelet transform to identify the isolated local maxima. The maxima mostly associated with impending earthquakes are then refined based on the geometrical continuity from spatial and temporal constraints to remove wrong maxima lines due to significant noise in input data. Wavelet maxima method is further utilized to analyze the result of eddy field method (Xiong et al., 2010).

Wavelet transform is based on the communication theory, thus its mathematical and physical foundation is complete, making it a highly promising method. The time series physical quantities derived from remote sensing are taken as the input information via a series of filtering and other processing, and finally thermal anomalies are extracted. Wavelet transform can decompose the thermal radiation into mutually independent frequency components almost without loss of information. Each component has a different dominant factor, thereby making the physical meaning much clearer than other methods. However,

the determination of high- and low-frequency signals in this method requires the geography- and geology-related knowledge with a certain arbitrariness. Although this algorithm achieves a high degree of decomposition and analysis of TIR signals, its mechanism that relates with earthquake preparation must to be further improved. Moreover, wavelet transform requires continuous time series of input data, but missing values in TIR data always appear due to cloud cover. Thus, the interpolation of missing data is required, which bring in uncertainties with regard to real values.





**3.5 Multiparameter comprehensive analysis**

Single precursor in pre-seismic monitoring research shows its limitations. Changes in various land, ocean and atmospheric parameters before and after strong earthquakes can be observed from satellite platforms, along with ground measurements. Comprehensive utilization of multiple precursors provides another promising choice, which can raise the reliability and credibility of the relationship between thermal anomalies and imminent seismic events (Eleftheriou et al., 2016; Singh et al.,

2010a). The preliminary step is to identify which precursors present anomalous variations prior to strong earthquakes around the epicenter. OLR, air temperature, and relative humidity were used in the seismic anomaly analysis for 2005 Mw 7.6 Kashmir and 2013 Mw 7.7 Awaran earthquakes (Mahmood et al., 2017). Singh et al. (2007) analyzed changes in SLHF, atmospheric water vapor, SST, wind, cloud liquid water, precipitation rate and chlorophyll concentrations prior to and after 2004 Sumatra earthquake. In short, many geophysical parameters present abnormal variations prior to earthquakes to some extent.

Anomalous variations of parameters in the surface, atmosphere and ionosphere prior to earthquakes present interrelations with one another and show the existence of strong coupling effects (Singh et al., 2007). Multiparametric analysis of the 2008 Wenchuan earthquake by Jing et al. (2013) indicates different response times for different parameters. Anomalous OLR occurred first, followed by air temperature, relative humidity, and air pressure presented abnormal variations. Finally, SLHF anomaly occurred the day before the main shock. Wu, *et al.* used multiple parameters from reanalysis datasets and ground

measurements to interpret the potential LCAC process (Wu et al., 2016). In Fig. 11, the precursor parameters of soil moisture, soil temperature, total column water vapour and screen-level air temperature present quasi-synchronous anomalies in the time window between 29 March and 1 April 2009 preceding the Mw 6.3 L'Aquila earthquake. Anomalous multiple parameters also indicate that the process of heat transfer is caused by tectonic movements, which is from the deep of crust to the land surface and then to the atmosphere. Multiparametric analysis helps in better understanding the sequence of processes that occurs in

the lithosphere, hydrosphere and atmosphere (Mansouri Daneshvar and Freund, 2017).

**3.6 Other methods**

The night thermal gradient (NTG) method calculates the linear trend of nighttime surface temperature using geostationary satellite LST data with high temporal resolution (Piroddi and Ranieri, 2012; Piroddi et al., 2014). In general, the surface temperature at night gradually decreases and reaches the minimum in the early morning, thus an NTG value greater than 0 is

regarded as an abnormal temperature increase. Furthermore, the most intensive anomalies of soil and air temperature are spatially coincident with TIR anomaly detected by NTG method (Wu et al., 2016). This approach is highly suitable for the geostationary satellite data that have continuous time series observations.

The interquartile method is used to establish upper and lower boundaries of temperature changes. The values that exceed boundaries are considered anomalous values that are associated with possible near-future earthquakes (Akhoondzadeh, 2013;





Saradjian and Akhoondzadeh, 2011). This method is only sensitive to the highest intensity anomaly, and its formula is as
follows:

$$M - k \cdot IQR \quad < V < M - k \cdot IQR, \tag{8}$$

where $M$ is the median value, $k$ is a threshold value, $IQR$ is the interquartile range between 75th and 25th percentiles, and $V$ is
the observed value. This formula can be reformed as

$$\left| \frac{V-M}{IQR} \right| < k. \tag{9}$$

The interquartile method is analogous to the ZS method, in which the average value and standard deviation are substituted by
the medium value and interquartile range, respectively.

   A variety of machine learning methods have been used in thermal anomaly detection, which show great advantages in
various fields in recent years and may have a considerable potential in this field. Autoregressive integrated moving average,

artificial neural network (ANN), support vector machine, and adaptive network-based fuzzy inference system (ANFIS) have
been compared for detecting thermal and total electron content anomalies (Akhoondzadeh, 2013). ANN model can handle well
the noise in input data. ANFIS that integrates both ANN and fuzzy inference systems is more effective at modeling non-linear
time series predictions than other methods. However, machine learning methods are always required to set certain empirical
parameters, and appropriate selection of these parameters is a challenging task in phases of model training and prediction. The

Kalman filter method is an iterative optimization system with high predictive power and is also used for detecting surface
temperature anomalies (Saradjian and Akhoondzadeh, 2011). This approach takes advantage of predicting nonlinear times
series data with non-normal distribution, such as the occurrence of thermal anomalies in LST data. These methods are more
complex and have not been widely used in thermal anomaly detection as other above mentioned methods. The comparison
among these methods with different independent properties should be conducted based on more earthquake cases in order to

screen out more effective approaches.

**4 Issues in thermal anomaly detection**

Considerable research work has long been done on possible seismic precursors in an apparent relationship with pre-seismic
complex tectonic activities. However, such studies are sometimes highly controversial, and the established correlation has
been considered with caution. The warming phenomena often occur prior to various earthquake cases, whereas the features of

these phenomena are often different. The interference of clouds is often highly serious, thereby resulting in faint and unstable
evolution patterns of warming areas, although certain cases show clear characteristics of thermal evolution. Even though many
hypotheses have been used to explain the mechanism of pre-seismic thermal anomalies, the physical mechanisms that drive
thermal anomalies remain unclear and are not widely accepted. Thus far, no objective, stable, and testable detection methods



or precursors are available for pre-seismic thermal anomalies. Consequently, thermal anomaly detection has not become a

well-recognized technology in the scientific community. At present, several problems remain in this domain, as detailed below.

1) Some important concerns of mentioned studies are the reasonable definition of "thermal anomaly" and quantitatively

measuring its quantity. Given that different definitions are used for different methods, data sources, or even earthquake cases,

one method based on one type of data source may get a good result for one earthquake, but it may be invalid when using these

standards in other earthquakes or the prediction. The arbitrary definition confuses anomalous signals by earthquake events and

impedes identifying abnormal thermal variations from complicated background signals.

2) Quantitatively validating results by anomaly detection methods is difficult partly because of the previous difficulty of the

definition. Whether thermal anomalies detected in one earthquake case are general or incidental phenomena is often

controversial. One of the general strategies is based on the validation and confutation method for verification (Genzano et al.,

2015; Lisi et al., 2010; Lisi et al., 2015; Xiong et al., 2013). In the validation phase, thermal anomalies associated with

earthquakes should be found; however, no thermal anomalies should be confirmed in the confutation phase. Nevertheless, the

validation and confutation approach alone is not enough. Most research results are based on and limited to a handful of case

studies. Therefore, obtaining convincing conclusion from statistical results is difficult when earthquake cases are insufficient

in the statistics analysis (Pulinets and Dunajecka, 2007).

3) Providing complete spatiotemporal data is difficult because of atmospheric interference, cloud cover, and instrument

performance. Satellite remote sensing on polar-orbiting or geostationary platforms provides the most important data sources

at regional or global scales, which are beneficial for global seismic monitoring studies. Surface data products, such as LST,

can be used in analyzing pre-seismic thermal anomalies to mitigate the atmospheric effect. However, the greatest weakness of

TIR remote sensing is that it cannot penetrate clouds to obtain surface thermal radiation information. This weakness results in

the existence of a large number of missing pixels in the data product, seriously affects thermal anomaly detection, hinders the

understanding of spatiotemporal evolution of thermal anomalies in a broader time scale, and finally leads to many cases of

omission or false positive rates (Lisi et al., 2015). Therefore, addressing the significant issue of cloud interference is critical in

thermal anomaly detection.

4) Other factors, including the surface (e.g., topography, geomorphology, and land covers) and atmosphere (e.g., cloud,

precipitation, and wind), also strongly influence the change of surface thermal radiation. Satellite thermal anomalies are

vulnerable to local atmospheric conditions, such as the inversion of air temperature, thereby resulting in inaccuracies in

detecting earthquake anomalies. Thus, more attention should be given to the possibility that other causes other than seismic

activities should be responsible for the extracted thermal anomalies. Especially, anthropogenic activities in factories and urban

areas also have a significant effect on thermal anomaly detection (Piroddi and Ranieri, 2012). Removing the normal





fluctuations of thermal emission related to meteorological or artificial factors from the earthquake-imposed thermal anomalies

is an urgent problem that must be solved.

    5) Nowadays, the data with low spatial resolution have been successfully used in thermal anomaly detection, such as NOAA OLR data with 2.5° spatial resolution (Liebmann, 1996). The spatial scale of OLR anomaly from 1°×1° NOAA data is approximately 1000–2000 km from epicenters (Xiong and Shen, 2017), which is an unexpected result for the seismic activity monitoring. However, the use of satellite data with a high spatial resolution (e.g., 1 km) can provide richer and deeper

information. This way, the spatial location of thermal anomaly can be better correlated with seismic activity areas, which is more conducive in studying the characteristics of spatial thermal anomalies, rather than limited to single point or local areas. For example, high- or low-temperature bands occur before rock bursts, and the main ruptures may occur along these band areas and result in thermal anomalies; however, data with coarse resolution are difficult to accurately capture such active tectonic positions.

6) Mechanism of pre-seismic thermal anomalies is inconclusive; thus, anomalous variations of such parameter can be or not be related to preparatory phases of an imminent earthquake. This situation leads to unclear physical mechanism of detection algorithms and arbitrary explanations about results of thermal anomalies. These defects seriously restrict the application and popularization of this approach. Several mechanisms for generation of pre-seismic thermal anomalies detected by satellites have been proposed and are in discussion (Saraf et al., 2009; Tramutoli et al., 2013). A unified LAIC model is proposed to

explain this phenomenon (Molchanov et al., 2004; Pulinets and Ouzounov, 2011). Wu, *et al.* added the coversphere to the LAIC model after analyzing its importance in the understanding of mechanisms and geophysical processes in earthquake preparation areas (Wu et al., 2016). However, further validation of these models is required from physical simulation experiences and synergetic measurements of multiparameter.

**5 Future development and perspectives**

Thus far, no single measurable geophysical variable and data analysis method illustrates considerable potential for a sufficiently reliable and effective operational earthquake forecasting practice. Forecasting moderate and strong earthquakes in the future several months to days at the regional or global scale is of high social value, which is the advantage for satellite thermal monitoring. Based on the discussion in the preceding section, the following perspectives are expected:

    1) Passive microwave remote sensing can obtain surface brightness temperature data under clouds, which is less affected by

atmospheric conditions. Although passive microwave data generally have lower spatial resolution (10–50 km in the nadir view) than that of TIR data (1–5 km), they remain sufficient for monitoring thermal anomalies at a global scale (Lisi et al., 2015). The fusion of TIR and passive microwave data is an important aspect for realizing the global observation of surface temperature (Duan et al., 2017; Wang et al., 2014). The data fusion technique can take advantage of the high precision and spatial resolution



of satellite TIR data and the penetrating clouds ability of microwave data. The improvement of spatial resolution of prospective

passive microwave sensors could be favorable to matching different data sources.

2) No reliable precursors have been found; thus, joint application and analysis of multiple parameters could be a choice. The results of multiparameter anomaly detection are combined to form a more reliable collective result, which reduces false thermal anomalies and uncertainties and improves the stability and accuracy of earthquake prediction. At the same time, ground measurements, such as surface and air temperatures along the most populous and active faults can provide a better

understanding and supplement the missing data of satellite imageries due to persistent cloud cover. Furthermore, other geophysical data, such as underground temperature, geomagnetism, gravity, properties of groundwater, and gases, can be combined with satellite data from the local, regional to global scales. This integrated study from different levels of the space, aerial platform, terrestrial surface, and underground will promote pre-seismic monitoring research (Venkatanathan et al., 2017).

3) A global-scale monitoring system based on a number of TIR data products, from polar-orbiting and geostationary

satellites, can offer complete earthquake cases analysis that render a better solution for detecting and analyzing the signals prior to major earthquakes. Furthermore, abundant satellite data, which are global long time series of continuous measurements, can support the sufficient long-term correlation research for various candidate precursors. Eleftheriou, *et al.* (Eleftheriou et al., 2016) analyzed M ≥ 4 earthquakes using OLR data in Greece for 10 years. Xiong and Shen (2017) also performed preliminary work on such global statistics of pre-seismic thermal anomalies. The statistical uncertainties can serve as a priori knowledge

to determine weight coefficients of precursors in a multiparametric monitoring system.

4) In the near future, remote sensing data, which are applicable to seismic activity monitoring, will be further increased with higher spatial, temporal or spectral resolutions. Thus, emerging technologies can be integrated to improve the predictive efficacy. The ultrahigh spectral TIR data could be used to derive surface temperature, emissivity, atmospheric profiles of temperature and humidity, and concentrations of gas composition simultaneously. Applying these parameters in joint

monitoring of seismic activity is a possible development direction. New satellite project, such as TwinSat, which integrates multiple observation technologies and ground measurements, can improve the understanding of seismic-triggered LAIC mechanism (Chmyrev et al., 2013).

5) Several methods can be synthetically applied to improve the efficacy of thermal anomaly detection. Because utilizing only a single approach to identify the pre-earthquake anomaly is not statistically robust (Bhardwaj et al., 2017). And integrating

multiple methods has shown good potentialities in improving the capability of detecting actual thermal anomalies. Given that qualitative analysis is subjectively influenced, developing quantitative anomaly detection method is a promising direction, which is also beneficial in realizing the comprehensive application of various methods. (Akhoondzadeh, 2013; Saradjian and Akhoondzadeh, 2011). For example, probability models that can integrate multiple methods or even various precursor variables may provide a more credible prediction with quantitative statistical uncertainties, other than the qualitative voting selection.



Meanwhile, developing new algorithms based on novel data mining technique (e.g., machine learning), which explore the

possibility of enhancing the performance in earthquake prediction, can optimize this multimethod scheme.

6) The evaluation of anomaly detection algorithms can contribute to investigate the defects of methods and verify their

validity. Building the quantitative determination of indicators and anomaly-identifying standards in the evaluation is

imperative, which is the basis for quantitatively estimating the efficacy of methods and comparability among them. According

to these indicators and standards, the big data statistical analysis based on long-term series of remote sensing data and global

earthquake cases is an important means to establish the statistical relationship between thermal anomalies and earthquake

events. In view of mathematics, these steps can maximally eliminate subjectivity. Furthermore, the credibility of satellite

thermal anomalies can be improved only after several means of verification and in-depth reveal of the regularity and precursory

characteristics of the earthquake. Analyses of detection results should combine with fault activities, crustal stress changes, and

underground heat flow. The spatial distribution of thermal anomaly should be consistent with the fault activity mode and its

mechanical model. After excluding the influence of non-tectonic factors, the derived thermal anomalies should be analyzed

together with measurements from other means (e.g., meteorological data). More importantly, the proposed methods could be

easily inspected by other researchers and ensure the reproducibility, thereby increasing statistical significance of detection

results. Thus, anomaly detection criteria should be clearly established and strictly used in other earthquake case studies and

subsequent earthquake prediction (Zhang et al., 2013).

7) The study of geophysical mechanisms and development of theoretical models about pre-seismic thermal anomalies should

be strengthened. The numerical simulation based on knowledge of seismo-tectonics can be used to establish the relationship

between TIR anomalous signals and seismic events. The diagnostic index with practical value could be created based on this

relationship, and the problem of anomaly index construction may be theoretically solved. The synergistic observations of

relevant parameters from underground to ionosphere in seismically active regions are necessary to validate these theoretical

models.

Yet still, further efforts on satellite thermal anomalies are required to study the physical mechanism, new satellite data, and

anomaly detection approaches. Although various techniques with different measurements and approaches have been used in

this field for decades and show the potential and possibility for the recognition of seismic precursors, no any technique has the

full ability to forecast imminent strong earthquakes in a short-term (e.g., a few weeks) or long-term (e.g., several years) for

now. Despite the difficulties in pre-seismic thermal monitoring, previous studies have shown that to some extent, a link exists

between thermal anomalies and seismic activities. Therefore, the practicality of thermal anomalies in earthquake monitoring

could be advanced from many perspectives based on existing research foundation. The anomalies induced by seismic activities

should be contained within the temporal and spatial distributions of satellite TIR information. Analyzing the correlation




between thermal anomalies and seismicity provides a promising way for short-term earthquake prediction and accelerates the

present capability on seismic hazard assessment and early warning.

**Author contribution:** Zhong-Hu Jiao designed the work and wrote the manuscript. Jing Zhao and Xinjian Shan modified the

paper and gave useful comments and suggestions.

**Acknowledgments:** This work is founded by Basic Science Research Plan of Institute of Geology, China Earthquake

Administration (Grant No. IGCEA1620) and Scientific Research Project of Shanghai Science and Technology Commission

(Grant No. 14231202600).

**Conflicts of Interest:** The authors declare no conflict of interest.

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



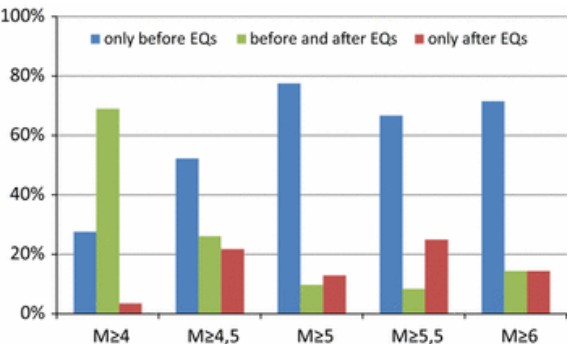

Figure 1. Histogram of thermal anomalies that occurred before and / or after earthquakes with different magnitudes

(Eleftheriou et al., 2016).

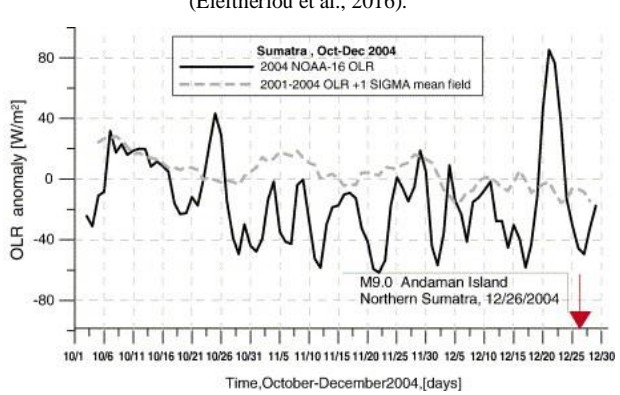

Figure 2. Daily OLR anomaly over the epicenter for Mw 9.0 Sumatra–Andaman Islands earthquake (Ouzounov et al.,

2007).



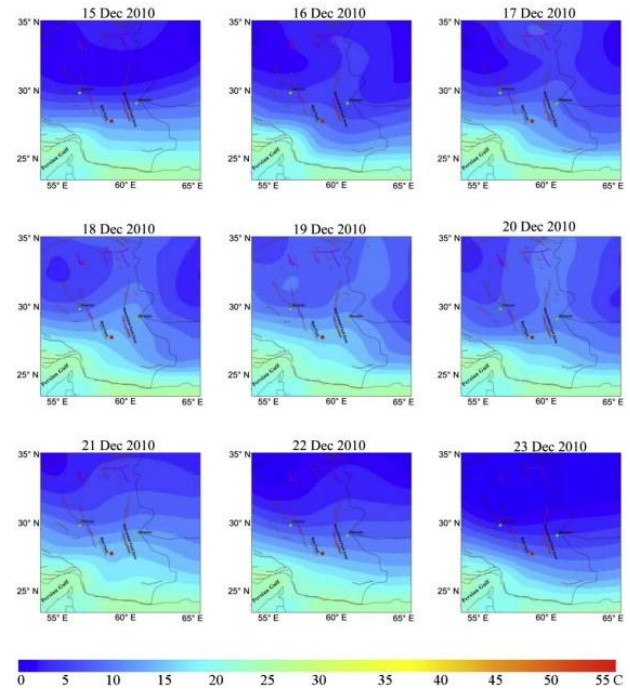

Figure 3. The variation of mean screen-level air temperature around the epicenter before and after the Ms 6.0 Kerman

earthquake (Alvan et al., 2013).

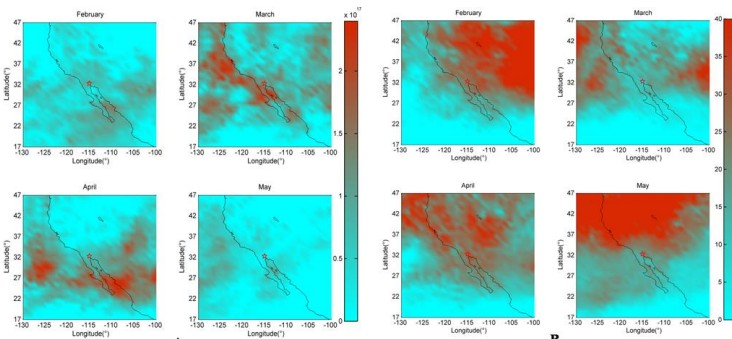

Figure 4. The anomalies of CO (mole cm$^{-2}$) (A) and O$_3$ (DU) (B), and the red star indicates the epicenter of the 2010 Baja

California earthquake (Cui et al., 2013).




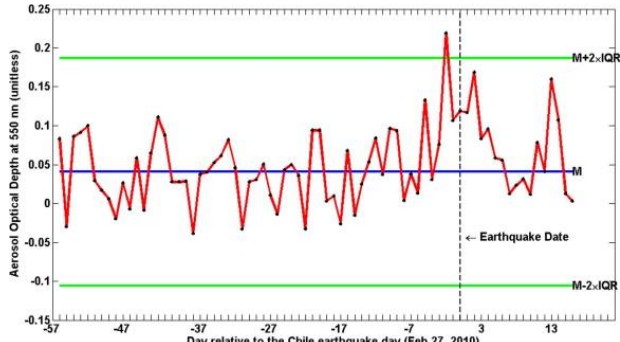

Figure 5. Time series of AOD anomalous values over the epicenter before the Mw 8.8 Chile earthquake at 27 February,

2010 (Akhoondzadeh, 2015). The vertical dotted line indicates the time of Chile earthquake, and median and upper / lower

bounds are the blue and green lines, respectively.

Figure 6. Time series of SLHF on the epicentre of the 2007 Ms 6.4 Pu'er earthquake (Qin et al., 2014a).


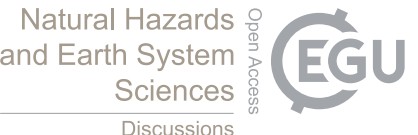

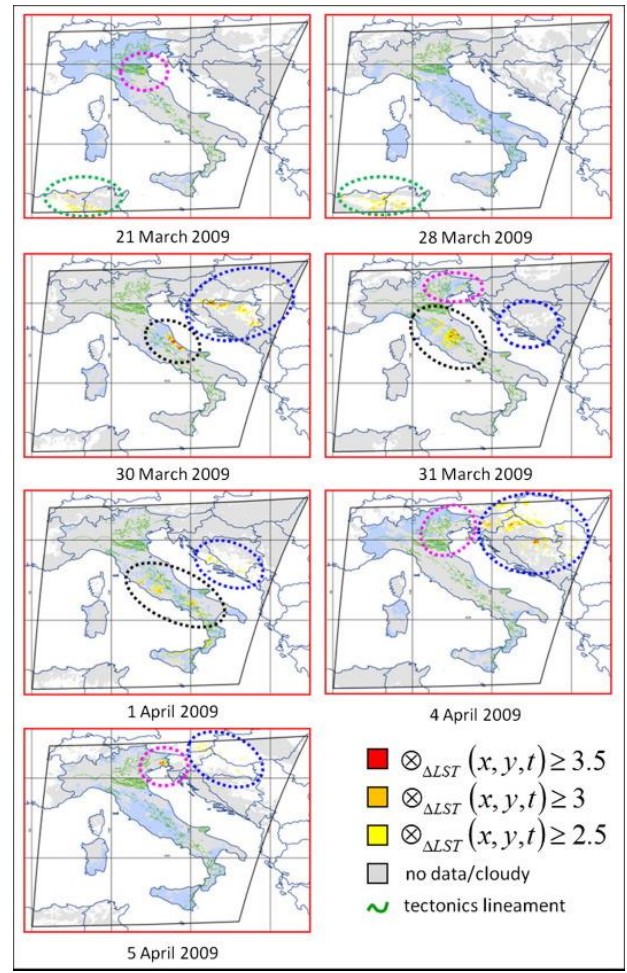

Figure 7. The map of LST anomalies indicated by colored dashed circles over the study area before the Mw 6.3 Aquila

earthquake on 6 April, 2009 (Lisi et al., 2015).

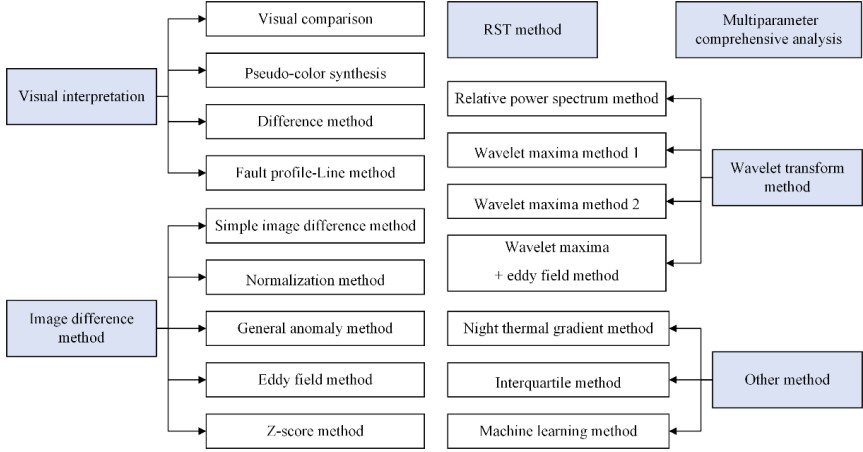





Figure 8. Pre-seismic thermal anomaly detection methods. The blue boxes indicate six classes that discussed below, and white boxes are the subclasses of each method.

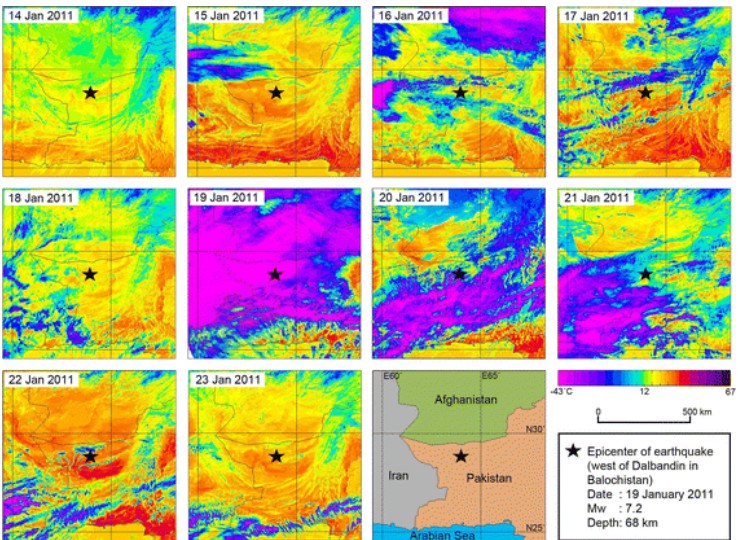

Figure 9. Time series of MODIS LST for the Dalbandin earthquake on January 19, 2011 (Saraf et al., 2012).

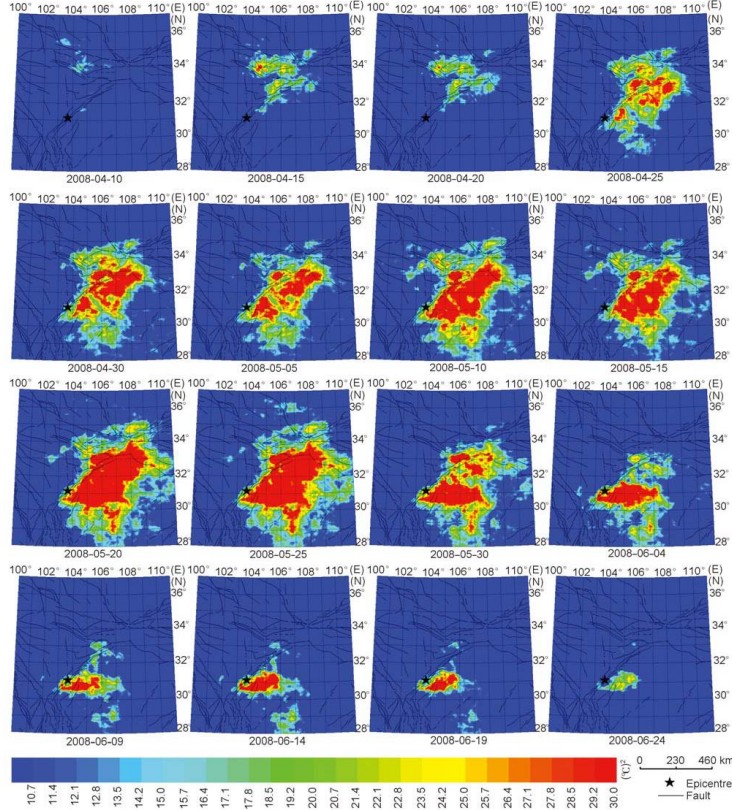



Figure 10. The spatio-temporal evolution of brightness temperature anomalies derived by the RPS method for the 2008

Ms8.0 Wenchuan earthquake (Zhang et al., 2010).

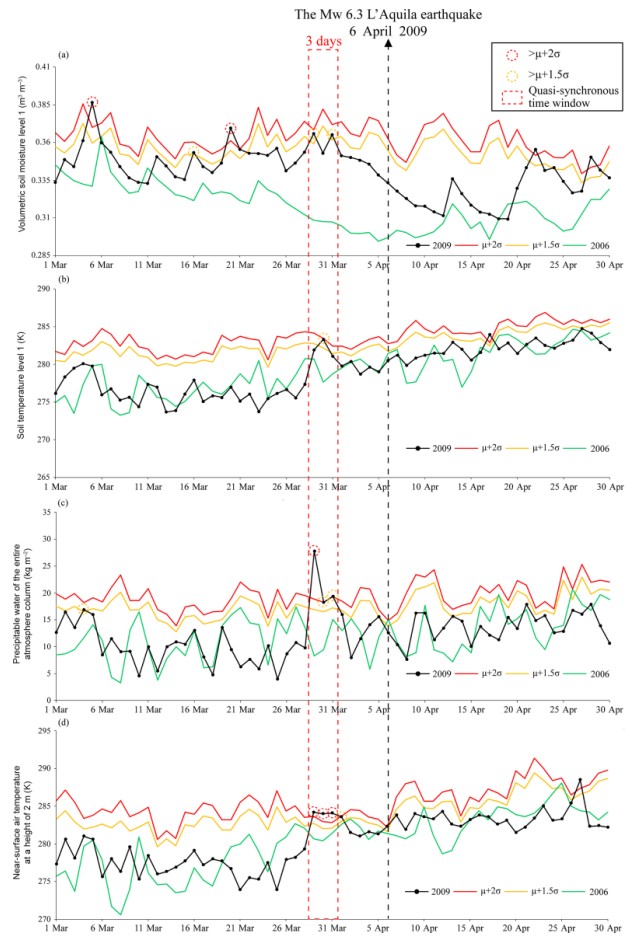

Figure 11. Time series of four seismic precursor parameters at the epicentral pixel from March to April 2009: volumetric

soil moisture level 1 (a), soil temperature level 1 (b), precipitable water of the entire atmosphere column (c), and near-

surface air temperature at a height of 2-meter (d) (Wu et al., 2016).

Table 1. The precursor parameters discussed in following sections

| Level | Parameter | Specific variables | Unit |
|---|---|---|---|
| TOA | brightness temperature | – | K |
| | OLR | – | W/m² |
| in the atmosphere | atmospheric water vapor | total column water vapor | cm |
| | atmospheric temperature | screen-level air temperature | K |
| | atmospheric trace gases | $CH_4$, $CO_2$, CO and $O_3$ | – |
| | aerosols | aerosol optical thickness | – |
| at the Earth's surface | SLHF | – | W/m² |

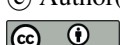



| | | |
|---|:---:|:---:|
| land surface temperature | – | K |
| sea surface temperature | – | K |

Note: the en dash ( – ) indicates no specific variables or unit.