# Peer review of "Pre-seismic Thermal Anomalies from Satellite Observations: A Review"

_Natural Hazards and Earth System Sciences, 2017_

## Referee Comment (RC1) · Dr. Freund (Referee) · 18 Oct 2017

On the surface, this is a fine review paper with a great deal of information presented in a comprehensive way, obviously with the intention to provide an overview of the most widely quoted interpretations of pre-earthquake thermal infrared anomalies.

What is the role of peer review? Is it to check whether the authors have well covered their chosen topic according to majority consensus or is it to check whether the authors have/have not penetrated through the thicket of misunderstanding that is rampant in their discipline. My evaluation of this review paper is primarily a critique that points at some very widespread and common misunderstandings in the Remote Sensing community when it comes to signals that the Earth emits prior to major earthquakes.

[Figure]

Since the authors cite one of my papers prominently at the beginning (Freund, F.: Pre-earthquake signals: Underlying physical processes, J. Asian Earth Sci., 41, 383-400, 2011), I feel compelled to point to some shortcomings of this manuscript – serious from my perspective. For me the question arises:

In the Introduction, on L 27-33, the text reads:

"Tectonic earthquakes are caused by the sudden dislocation of active faults due to surging tectonic stress (Freund, 2011). In addition to the considerable amount of strain energy released during the earthquake itself, the stress energy continuously accumulates during the preparation process of the earthquake. The activity of faults in earthquake-prone areas often results in the growth of surface microfissures and gas ionization effects, following with changes in water content, underground gas, and earth electro-magnetism around active faults. To some extent, these changes lead to pre-seismic thermal anomalies in seismogenic areas, such as regional warming and increased greenhouse gas concentration, which can be observed through satellite sensors."

In my cited 2011 publication I had presented at great length that the most important processes during the earthquake preparation process are (1) the build-up of tectonic stresses and (2) the activation of omnipresent defects in all crustal rocks, which releases electronic charge carriers, called "positive holes". (1) is obvious and accepted by everybody. (2) is based on a large body of work that I have published since the 1980s, first in a basic material sciences context, unrelated to earthquakes, but subsequently applied to earthquakes and, specifically, pre-earthquake processes.

Though the authors of this Review cite my 2011 paper, they seem to have missed or misunderstood its contents. This is obvious from their list of follow-on processes to which the authors draw attention, namely "…the growth of surface microfissures and gas ionization effects, following with changes in water content, underground gas, and earth electromagnetism around active faults… regional warming and increased greenhouse gas concentration," This list indicates that the authors are intent on reviewing

the relevant literature without taking into account my work. Why, then, do they cite my 2011 paper at such a prominent place?

Frankly, despite the near-universally cited 1968 BSSA paper by Chris Scholz "Microfractures, aftershocks, and seismicity", nobody has ever presented evidence that microfracturing is taking place in the Earth crust, either at the surface, in shallow depth or at great depth. Nobody seems to have ever raised the question, whether it is possible for rocks at seismogenic depth (7-45 km and deeper) to undergo microfracturing. I mind you, every fracture event, micro or macro, is possible only, if the volume can expand. The reason is that, by definition, fracturing creates new surfaces. Creating new surfaces is possible only if and when empty space is created between the two sides of the crack. However, at the depth of kilometers to tens of kilometer, the overload of the rock column is such that the amount of work to be done (thermodynamically) to increase the volume of the stressed rocks is very large. Hence, the chance of creating any fracturing, micro or macro, is very small. Nonetheless the geoscience community, including the authors of this review, blankly accept the microfracturing maxim.

If one digs deeper into why microfracturing is so popular, an interesting story emerges. Geophysicists have for decades noted increases in the electrical conductivity of the rock volumes deep in the crust that are being stressed prior to major seismic events. Nobody could explain such increases except by assuming that brines were penetrating into the stressed rock volumes. Hence, the assumption that fractures must be opening deep below allowing water to rush in. This facile explanation was so tempting that nobody seems to notice that this assumption contradicts the fact that, below 5-7 km depth, the open porosity of rocks disappears. The reason: the difference between hydrostatic and lithostatic pressure becomes so large that open porosity cannot be maintained – even not over geologically short time scales.

Throwing in the words "gas ionization effects" reinforces the impression that the authors have not made an effort to inform themselves about HOW air ionization at the Earth surface takes place. Likewise, what do they mean by writing "with changes in

water content, underground gas (and) earth electromagnetism around active faults"? These are meaningless words unless substantiated by some physical insight into the underlying processes. I have gone to some length describing the underlying physical processes in my 2011 paper, including electromagnetic processes and air ionization at the Earth surface. I have the impression that the authors of this review have not made an effort to familiarize themselves with these processes.

I can go on with my critic (which I offer in a constructive spirit) when I read in L49 "due to the unclear physical mechanism of pre-seismic thermal anomalies". I for one posit that the physical mechanisms are no longer "unclear". The authors' misconception comes from the fact that they don't realize the difference between "thermal anomalies" and "thermal infrared anomalies". The difference is huge– from a physics perspective. Saying "thermal anomalies" automatically implies a temperature difference, e.g. a "tangible" Joule temperature difference. Saying "thermal infrared anomalies" refers to the ONLY observables that infrared-sensing satellites can deliver: intensity and, to some extent, spectral distribution of the infrared emitted from the ground, from the lower atmosphere and from the top of the atmosphere. All of remote sensing depends upon the interpretation of these infrared emission processes.

Much of the remaining paper endorses, either implicitly or directly, the conventional interpretation of the different kinds of remotely sensed pre-earthquake IR anomalies. I'm convinced that the remote sensing community has been on the wrong track for most of the time, I but hesitate to express my concerns. The reason is that my concerns are so fundamental that, if rigorously applied, not much is left of this review paper to recommend. However, I want to help the authors.

For instance, on L428, late in their Review, under 3.6 Other methods, they introduce the night thermal gradient (NTG) method, first used by Nevin Bryant at JPL and then applied extensively by Luca Piroddi and Gaetano Ranieri in Italy as quoted in L430. Regrettably, the authors continue to use the blanket terminology "surface, soil and air temperature" without mentioning that they are actually talking about the "radiative

temperature" derived from infrared emission off the surface, the soil and the air.

Why is the NTG method introduced so late in this review and under the title Other methods? The authors do not realize that, by using data from the European geostationary satellite (providing thermal images every 15 min) Piroddi's work has provided much more profound information. For instance, by analyzing a full year of night-time data for the entire Italian peninsula, Piroddi has shown (1) that regional TIR anomalies come and go over the course of time, in a matter of days, expanding over relatively wide areas, but only occasionally linkable to large seismic events, (2) that the TIR intensities wax and wane on time scales of hours, (3) that the TIR anomalies move across the landscape on time scales even shorter than hours, and – most importantly – (4) that the TIR anomalies have a clear tendency to be associated with hill tops and mountain tops. In fact, the intensity of the TIR emissions from valleys is much less than from the tops of adjacent mountains. If the authors of this review paper would have paid more attention to the work by Piroddi and his thesis advisor, Professor Ranieri, they would have noted that the populist interpretation of the TIR anomalies off the Earth's surface, namely that they are due to warm gases or greenhouse gases seeping out of the ground, must be fundamentally wrong. The NTG analysis clearly points to an alternative mechanism, for which I have laid the groundwork: IR emission due to the radiative de-excitation of peroxy entities at the Earth surface. I attach an extended abstract from the 2015 EMSEV Workshop, in which the preference of the TIR emission from mountain tops is unambiguously documented (at least for one well studied case, the M=6.3 2009 L'Aquila event).

All this also links to the Section 4 Issues with thermal anomaly detection. It is correct, as the authors note in L460, that the issue is "highly controversial", but they do not penetrate the superficial appearance of the widespread controversy. In L461 they use the word "warming". The casual use of this word reveals that they= authors do not understand the physical principle of the radiative nature of the remote sensing signals analyzed by the community.

In L512-514 the authors refer to the "unified LAIC model, widely promoted by Sergey Pulinets and his numerous collaborators. However, a close examination of the LAIC model reveals that it is based on ad hoc assumptions regarding radon. Radon has been proposed to the driver of the LAIC model even though, in the larger context, it is physically impossible that radon can play this role. If radon were responsible to the increase of air ionization prior to major seismic events, it would have to increase the normal air ion concentration from the "fair weather average" of about 200 per cubic centimeter to 20,000 to 50,000 per ccm. In average crustal rocks, radon is rarer than gold by 6 orders of magnitude. There is about one mole Rn in Earth's atmosphere. Measured close to the ground or in holes in the ground, the pre-earthquake Rn emanation increases by a factor of about 10. Just calculate the number of Rn atoms per ccm of normal air and ask yourself, how the decay of these rare Rn atoms can cause a regional increase of the air conductivity by a factor 100-250.

In summary, this is a fine paper with lots of references, but it suffers from the fact that the authors do nothing but reinforce the mainstream conception that the question of the so-called "thermal anomalies" (which are in fact infrared intensity anomalies) is so complex that it cannot be solved. I disagree with this assessment. I regret that the authors have not been able to go beyond the simplistic and physically untenable explanations why there are changes in the IR emissivity of the Earth surface around earthquake preparation zones.

---

## Referee Comment (RC2) · M. F. Buongiorno (Referee) · 28 Oct 2017

The paper is well constructed and gives to readers a full view of the current status of studies regarding the , use of temperature anomalies for earthquake prediction Chapter 3,4 and 5. Nevertheless the chapter 2 need to be reviewed in order to give a full description of different parameters used to detect thermal anomalies related to seismic activity. Is misleading in this Chapter that the author indicates that satellite measurements have already proven to be able to detect thermal anomalies as precursors of seismic events (pag. 4, line 95-100) when this thesis is then refuted in Chapters 4 and 5 considering that the mechanism that produced the possible anomalies is not well understood and the measurements of temperature, gas and aerosol anomalies could not be related with enough confidence to seismic activity due also to instrumental limitation. Chapter 2 should remain the description of the possible usable parameters tritrieved by satellite and used in the algorithms and approaches described in chapter 3.

The author refers to a list of parameters which could be derived by satellite measurements but due to the large number of satellite data used to extract the presented parameters, I would suggest to add a table that shows the satellite missions, the possible retrieved parameter, parameter dimension, estimated accuracy and error, spatial and temporal resolution in order to permit the reader to understand the difficulties that those studies are encountering to extract significant measurements for the earthquake precursor analysis. Chapter 2 needs to be more coherent with the following Chapters that are giving a correct critical review of this fascinating research aimed to the understanding the complex interaction between Earth interior and surface phenomena. The author may consider focusing only on temperature anomalies excluding satellite retrievals of columnar gas content anomalies due to the emissions in the fault zones. I strongly believe that this is not a suitable parameter if satellite measurements are considered since current satellites measure $CO_2/CO$ columnar concentrations with very coarse spatial resolution and therefore with very low chance to detect appreciable variations on localized areas. Volcanic emitted $CO_2$ which shows high concentrations and is continuously emitted is mostly not detected by current satellite missions due to the high concentration in the atmosphere and the quick dispersion. Fault areas could discharge gases but the direct measurement of an anomalous concentration in the atmospheric column over the possible earthquake area by current satellites missions is not credible.

A final comment which may be added to the Chapter 5 is to strengthen the consideration that the studies on the analysis of thermal anomalies are of high scientific interest but at the moment this study could not bring any practical use for the earthquake risk reduction or alert. I appreciate the very clear statement on fact that we need to improve satellite instruments capabilities in terms of accuracy and spatial/temporal resolution. This review may stimulate the development of specific experiments which will help the

understand more about the interactions between faults movements and the variation of surface parameters which could be detected by satellites observations.

---

## Author Comment (AC1) · 9 Nov 2017

Dear Prof. Buongiorno,

Thank you very much for your careful checking of the manuscript and the insightful comments and suggestions. After detailed revisions, we think the paper has been improved according to your helpful suggestions. Our responses are also enclosed in the pdf file.

Q1:

The paper is well constructed and gives to readers a full view of the current status of studies regarding the , use of temperature anomalies for earthquake prediction Chapter 3,4 and 5. Nevertheless the chapter 2 need to be reviewed in order to give a full

description of different parameters used to detect thermal anomalies related to seismic activity. Is misleading in this Chapter that the author indicates that satellite measurements have already proven to be able to detect thermal anomalies as precursors of seismic events (pag. 4, line 95-100) when this thesis is then refuted in Chapters 4 and 5 considering that the mechanism that produced the possible anomalies is not well understood and the measurements of temperature, gas and aerosol anomalies could not be related with enough confidence to seismic activity due also to instrumental limitation. Chapter 2 should remain the description of the possible usable parameters retrieved by satellite and used in the algorithms and approaches described in chapter 3.

Answer:

Thanks for your concerns and suggestions. In fact, based on those relatively highly cited and accepted papers which were seriously selected from many good scientific journals, we aim to review the possible precursors for pre-earthquake anomaly detection in Section 2, as well as state several frequently used anomaly detection methods in Section 3. Although these precursors and methods give us some uplifting results in which we can notice some significant anomalies prior to many earthquake cases, we still need to take some criticism and deeper insights on them in order to achieve the further aim, i.e. forecasting moderate and strong earthquakes ahead. According to our forecast practice and research there are still many issues that we have to figure out in order to achieve a more robust prediction. For example, we can find some anomalies prior to an earthquake that already happened, but when some anomalies occur we cannot predict an impending earthquake very confidently. Accordingly, we humbly point out the defects or misunderstandings in the above stated studies from the definition to data sources and several other aspects in Section 4. Besides, we list the future progresses in this field in Section 5, considering the descriptions in Section 4.

In a word, the logic of this paper goes like, Section 2 and 3 state the current study

situation, then Section 4 and 5 analyze them and try to give some suggestions on the future development and perspectives. In order to make this logic more clearly, we also add some explanations at the beginning of Section 4 as follows.

"We review the possible precursors for pre-earthquake anomaly detection in Section 2, as well as state several frequently used anomaly detection methods in Section 3. Although these precursors and methods give us some uplifting results in which we can notice some significant anomalies prior to many earthquake cases, we still need to take some criticism and deeper insights on them in order to achieve the further aim, i.e. forecasting moderate and strong earthquakes ahead. Accordingly, in this section, we humbly point out the defects or misunderstandings in the above stated studies from definition to data sources and several other aspects. "

Q2:

The author refers to a list of parameters which could be derived by satellite measurements but due to the large number of satellite data used to extract the presented parameters, I would suggest to add a table that shows the satellite missions, the possible retrieved parameter, parameter dimension, estimated accuracy and error, spatial and temporal resolution in order to permit the reader to understand the difficulties that those studies are encountering to extract significant measurements for the earthquake precursor analysis.

Answer:

Thanks for your kind advice. We have added a more comprehensive table in company with Table 1. This table can be found in the attached pdf file. It contains several selected sensors which are in operation, have abundant and high-quality products, and are widely employed in remote sensing community.

Q3:

Chapter 2 needs to be more coherent with the following Chapters that are giving a correct critical review of this fascinating research aimed to the understanding the complex interaction between Earth interior and surface phenomena. The author may consider focusing only on temperature anomalies excluding satellite retrievals of columnar gas content anomalies due to the emissions in the fault zones. I strongly believe that this is not a suitable parameter if satellite measurements are considered since current satellites measure $CO_2$/CO columnar concentrations with very coarse spatial resolution and therefore with very low chance to detect appreciable variations on localized areas. Volcanic emitted $CO_2$ which shows high concentrations and is continuously emitted is mostly not detected by current satellite missions due to the high concentration in the atmosphere and the quick dispersion. Fault areas could discharge gases but the direct measurement of an anomalous concentration in the atmospheric column over the possible earthquake area by current satellites missions is not credible.

Answer:

Thanks for your concerns. We will explain the reasons one by one.

Firstly, as we said in the response to question 1, Section 2 selectively describes several pre-earthquake precursors which has been proposed in advance and then accepted by some other scientists. We aren't intended to introduce any novel precursor but to review the related parameters which will be further discussed in the following contents. And the credibility of trace gases require to be validated as we said in Section 4 and 5.

Secondly, the trace gases derived from satellite data always have low spatial resolution ( like 1°), therefore they cannot indicate the regional anomalous variations that should use satellite data with higher resolution. This situation is very similar to the usage of OLR data that also have low spatial resolution. From the point of view of many papers using satellite derived trace gases or OLR data, they present high relationship between the anomalous areas and the location of an earthquake. Therefore it seems that low spatial resolution is not a very serious problem. $CO_2$ and some other trace gases are believed to induce local greenhouse effects. Therefore, they are indeed thermal

anomaly related parameters.

Thirdly, central volcano even shield volcano is somehow like a point emission source, while its counterpart fault is always like a line or area source when considering their relative areas. Although the emission of volcano is more intensive, just as you said, its limited concentration area and quick dispersion can hardly be captured by satellites with the coarse spatial and temporal resolutions. As for trace gases which induced by fault movements, they are often long-lasting and large-existing, though with lower concentration. Meanwhile the temporal resolution of data acquirement can be 1 - 2 per day. The data can capture the variations of trace gases in every day, thus the transient changes of gas concentration may also be recorded. Therefore, they might be caught by some scientists, just as described in Section 2.3 "Atmospheric trace gases".

In the Section 5, we also point out that with the assistance of in-situ measurements in specific places. Ground measurements of trace gases are also used in the earthquake monitoring, and trace gases retrieved from remote sensing data can be considered as the extension of the ground measurements.

Last, the advance of sensor in the future could provide higher spatial resolution and accuracy. The remotely sensed trace gases data with higher spatiotemporal resolution and accuracy might provide more useful information for the earthquake prediction.

In conclusion, trace gases can be the possibility in earthquake monitoring, and could be a candidate precursor as other precursors for now despite with many limits that hinder the application of monitoring seismic activities.

Q4:

A final comment which may be added to the Chapter 5 is to strengthen the consideration that the studies on the analysis of thermal anomalies are of high scientiïÏňĄc interest but at the moment this study could not bring any practical use for the earthquake risk reduction or alert. I appreciate the very clear statement on fact that we need

to improve satellite instruments capabilities in terms of accuracy and spatial/temporal resolution. This review may stimulate the development of specific experiments which will help the understand more about the interactions between faults movements and the variation of surface parameters which could be detected by satellites observations.

Answer:

Thank you for your appreciation. As you suggested, we modified the last paragraph (in the red) in Section 5 as follows.

[revised manuscript text omitted]

Please also note the supplement to this comment:
https://www.nat-hazards-earth-syst-sci-discuss.net/nhess-2017-211/nhess-2017-211-AC1-supplement.pdf

---

## Author Comment (AC2) · 14 Nov 2017

Dear Dr. Freund,

Thank you very much for your careful checking of the manuscript and the insightful comments and suggestions. After detailed revisions, we think the paper has been improved a lot. Our responses to the comments and suggestions are also enclosed in the pdf file.

Q1.

In the Introduction, on L 27-33, the text reads:

"Tectonic earthquakes are caused by the sudden dislocation of active faults due to

surging tectonic stress (Freund, 2011). In addition to the considerable amount of strain energy released during the earthquake itself, the stress energy continuously accumulates during the preparation process of the earthquake. To some extent, these changes lead to pre-seismic thermal anomalies in seismogenic areas, such as regional warming and increased greenhouse gas concentration, which can be observed through satellite sensors." In my cited 2011 publication I had presented at great length that the most important processes during the earthquake preparation process are (1) the build-up of tectonic stresses and (2) the activation of omnipresent defects in all crustal rocks, which releases electronic charge carriers, called "positive holes". (1) is obvious and accepted by everybody. (2) is based on a large body of work that I have published since the 1980s, first in a basic material sciences context, unrelated to earthquakes, but subsequently applied to earthquakes and, specifically, pre-earthquake processes.

Though the authors of this Review cite my 2011 paper, they seem to have missed or misunderstood its contents. This is obvious from their list of follow-on processes to which the authors draw attention, namely ". . .the growth of surface microfissures and gas ionization effects, following with changes in water content, underground gas, and earth electromagnetism around active faults. . . regional warming and increased greenhouse gas concentration," This list indicates that the authors are intent on reviewing the relevant literature without taking into account my work. Why, then, do they cite my 2011 paper at such a prominent place?

Throwing in the words "gas ionization effects" reinforces the impression that the authors have not made an effort to inform themselves about HOW air ionization at the Earth surface takes place. Likewise, what do they mean by writing "with changes in water content, underground gas (and) earth electromagnetism around active faults"? These are meaningless words unless substantiated by some physical insight into the underlying processes. I have gone to some length describing the underlying physical processes in my 2011 paper, including electromagnetic processes and air ionization at the Earth surface. I have the impression that the authors of this review have not made

an effort to familiarize themselves with these processes. '

Answer:

In this paper, we attentively review the advances in possible earthquake precursors and thermal anomaly detection approaches over the last decade. Thus, we did not give insight into various literatures about the mechanisms of pre-seismic anomalies. We just simply described the possible physical processes that we had known from the papers we read routinely, resulting in some misunderstanding about different theories or mechanisms. According to your suggestions and criticism, we referred many relevant papers and rewrote the mentioned paragraph as follows:

Tectonic earthquakes are caused by the sudden dislocation of active faults due to surging tectonic stress. In addition to the considerable amount of the strain energy released during the earthquake itself, the stress energy continuously accumulates during the preparation process of the earthquake. Different theories to explain the physical mechanism of the pre-seismic anomalies derived from optical satellite data have been proposed. The p-hole model (Freund, 2011; Freund et al., 2009) indicates that electronic charge carriers, also known as positive holes, in crustal rocks activated by tectonic stress and flow out of the stressed rock volume and propagate fleetly. They cause the air ionization at the land surface-atmosphere interface when accumulated in a thin surface/subsurface layer, and generate non-thermal infrared emission as a result of the recombination of positive holes. The rock stress adjustment from active faults probably causes anomalies of land surface or air temperatures prior to the earthquake, which could be observed through satellite TIR sensors (Chen et al., 2015; Ren et al., 2017; Wu et al., 2006). The spatiotemporal evolution of rock temperature field is closely related with its deformation. The rock shear strain or compression causes the obvious increase of temperature, and rock tension gives rise to temperature reduction. The solid Earth is about 1650 times of thermal capacity of the atmosphere. The change of elastic stress of 1 MPa is likely to bring in the variation of air temperature with the order of 1 K based on the energy balance. The local greenhouse effect due to the emanation of CO2, CH4, etc. has been invoked to explain the anomalous variations of TOA brightness temperature or OLR (Ouzounov et al., 2006; Ouzounov et al., 2007; Tronin et al., 2002). Besides, the increased emission of radon from active faults and cracks in seismogenic regions is also considered to bring about the air ionization, which can concentrate water molecules on air ions, further lead to the anomalies of the atmospheric water vapor and temperature, and accelerate the latent heat flux before earthquakes (Pulinets et al., 2006).

Q2.

Frankly, despite the near-universally cited 1968 BSSA paper by Chris Scholz "Microfractures, aftershocks, and seismicity", nobody has ever presented evidence that microfracturing is taking place in the Earth crust, either at the surface, in shallow depth or at great depth. Nobody seems to have ever raised the question, whether it is possible for rocks at seismogenic depth (7-45 km and deeper) to undergo microfracturing. I mind you, every fracture event, micro or macro, is possible only, if the volume can expand. The reason is that, by defi̧nition, fracturing creates new surfaces. Creating new surfaces is possible only if and when empty space is created between the two sides of the crack. However, at the depth of kilometers to tens of kilometer, the overload of the rock column is such that the amount of work to be done (thermodynamically) to increase the volume of the stressed rocks is very large. Hence, the chance of creating any fracturing, micro or macro, is very small. Nonetheless the geoscience community, including the authors of this review, blankly accept the microfracturing maxim.

If one digs deeper into why microfracturing is so popular, an interesting story emerges. Geophysicists have for decades noted increases in the electrical conductivity of the rock volumes deep in the crust that are being stressed prior to major seismic events. Nobody could explain such increases except by assuming that brines were penetrating into the stressed rock volumes. Hence, the assumption that fractures must be opening deep below allowing water to rush in. This facile explanation was so tempting that nobody seems to notice that this assumption contradicts the fact that, below 5-7 km

depth, the open porosity of rocks disappears. The reason: the difference between hydrostatic and lithostatic pressure becomes so large that open porosity cannot be maintained – even not over geologically short time scales.

Answer :

Thanks for the detailed comments about the microfracturing. Frankly speaking ,we did take the microfracturing at the seismogenic depth (7-45km and deeper) for granted, which might resulting from our major, i.e. remote sensing, as well as the relative lack of the fundamental knowledge of seismology and geology. Your explanation enlightened us and made us think about the theory in a new way. To remind the readers who might have similar misunderstanding or ignorance, we learned from your explanation and modified the item 7) in the Section 5 as follows.

7) The study of geophysical mechanisms and development of theoretical models about pre-seismic thermal anomalies should be strengthened or even updated. The numerical simulation based on knowledge of seismo-tectonics can be used to establish the relationship between anomalous signals and seismic events. The diagnostic index with practical value could be created based on this relationship, and the problem of anomaly index construction may be theoretically solved. Meanwhile, the former theories should also be examined with new minds and technologies. For example, the microfracturing theory is somehow fragile (Freund, 2011). By the basic definition, fracturing can create new surfaces and is possible only if empty space is created between the two sides of the crack. For the rocks at seismogenic depth (7-45 km and deeper), the overload is so large that the chance of creating any micro- or macro-fracturing, is very small. In other word, the difference between hydrostatic and lithostatic pressure becomes so large that open porosity of rocks cannot be maintained. Besides, the synergistic observations of relevant parameters from underground to ionosphere in seismically active regions are necessary to validate these theoretical models.

Q3.

[Figure]

I can go on with my critic (which I offer in a constructive spirit) when I read in L49 "due to the unclear physical mechanism of pre-seismic thermal anomalies". I for one posit that the physical mechanisms are no longer "unclear". The authors' misconception comes from the fact that they don't realize the difference between "thermal anomalies" and "thermal infrared anomalies". The difference is huge– from a physics perspective. Saying "thermal anomalies" automatically implies a temperature difference, e.g. a "tangible" Joule temperature difference. Saying "thermal infrared anomalies" refers to the ONLY observables that infrared-sensing satellites can deliver: intensity and, to some extent, spectral distribution of the infrared emitted from the ground, from the lower atmosphere and from the top of the atmosphere. All of remote sensing depends upon the interpretation of these infrared emission processes.

Much of the remaining paper endorses, either implicitly or directly, the conventional interpretation of the different kinds of remotely sensed pre-earthquake IR anomalies. I'm convinced that the remote sensing community has been on the wrong track for most of the time, I but hesitate to express my concerns. The reason is that my concerns are so fundamental that, if rigorously applied, not much is left of this review paper to recommend. However, I want to help the authors.

For instance, on L428, late in their Review, under 3.6 Other methods, they introduce the night thermal gradient (NTG) method, first used by Nevin Bryant at JPL and then applied extensively by Luca Piroddi and Gaetano Ranieri in Italy as quoted in L430. Regrettably, the authors continue to use the blanket terminology "surface, soil and air temperature" without mentioning that they are actually talking about the "radiative temperature" derived from infrared emission off the surface, the soil and the air.

Answer :

Thanks for your kindness. We did have some misunderstanding about "thermal anomalies". We misused "thermal anomalies" as a general concept that indicate all the anomalies of thermal radiation related parameters, such as outgoing longwave radiation, water vapor and land surface temperature (LST). Meanwhile "thermal infrared anomalies" are considered as the anomalies of TOA radiances or brightness temperature, which include the thermal infrared (TIR) emissions from the ground surface and the entire atmosphere.

The parameters derived from satellite data are different from the data itself. Various parameters can be retrieved from the multispectral optical satellite data as mentioned in our paper. For example, the LST can be retrieved from two thermal infrared atmospheric window channels. Although LST is derived from TIR data, but it is no longer remotely sensed TIR radiance. The LST represents a remarkably thin surface layer of medium temperature state, which is a physical quantity that can also be measured at the ground. However, the retrieved LST is not exactly same as the ground measurements, and the accuracy is used to express this bias.

In order to clearly and simply express the concept of the "pre-earthquake anomalies", we update the expression as "LST anomaly", "water vapor anomaly" or "OLR anomaly" instead of calling them "thermal anomalies" generally and ambiguously. Besides, the title of this paper is also modified as "Pre-seismic Anomalies from Optical Satellite Observations: A Review".

After reading more papers about the physical mechanism of pre-seismic anomalies, I agree with you that the physical mechanism is already relatively clear and partially proven. However, we have to admit that because of the complexity of seismogeology and geophysics, various theories and mechanisms have not been widely verified and accepted. As for more detailed application logic of remote sensing technology in the field of the pre-earthquake anomalies, as well as the thoughts on the present study track, we would like to make more effort in the next paper that focuses on our method.

Q4.

Why is the NTG method introduced so late in this review and under the title Other methods? The authors do not realize that, by using data from the European geostationary

satellite (providing thermal images every 15 min) Piroddi's work has provided much more profound information. For instance, by analyzing a full year of night-time data for the entire Italian peninsula, Piroddi has shown (1) that regional TIR anomalies come and go over the course of time, in a matter of days, expanding over relatively wide areas, but only occasionally linkable to large seismic events, (2) that the TIR intensities wax and wane on time scales of hours, (3) that the TIR anomalies move across the landscape on time scales even shorter than hours, and – most importantly – (4) that the TIR anomalies have a clear tendency to be associated with hill tops and mountain tops. In fact, the intensity of the TIR emissions from valleys is much less than from the tops of adjacent mountains. If the authors of this review paper would have paid more attention to the work by Piroddi and his thesis advisor, Professor Ranieri, they would have noted that the populist interpretation of the TIR anomalies off the Earth's surface, namely that they are due to warm gases or greenhouse gases seeping out of the ground, must be fundamentally wrong. The NTG analysis clearly points to an alternative mechanism, for which I have laid the groundwork: IR emission due to the radiative de-excitation of peroxy entities at the Earth surface. I attach an extended abstract from the 2015 EMSEV Workshop, in which the preference of the TIR emission from mountain tops is unambiguously documented (at least for one well studied case, the M=6.3 2009 L'Aquila event).

Answer:

We agree that the NTG method has relative clearly physical definition and is effective for the anomaly detection. Nevertheless, we have to admit that it has not been widely used or frequently referred in present scientific papers. In this paper, we intend to list and discuss the possible seismic precursors and detection methods selectively based on their respective application and influence. Thus, we put the NTG method under the section "Other methods". As you said, "only occasionally linkable to large seismic events" is an important reason why we discuss the limits of this technique in Section 4 and 5. We also point out the drawback of the TOA brightness temperature, which is

effected strongly by the entire atmosphere in Section 2.2. It is also one of the reasons that cause the variation of TIR intensities at a short time scale. The terrain effects that indicated by the NTG method might be related with the fact that this method does not remove the TIR background information from current observations, which is an essential step in the Z-score or RST methods. Of course, it can also be explained by the alternative mechanism of IR emission due to the radiative de-excitation of peroxy entities at the Earth surface.

We would love to learn some merits from the 2015 EMSEV Workshop abstract that you mentioned above. However, we did not find it in the attachment and failed to search it on the Internet. Could you please offer us more information about it?

Q5.

All this also links to the Section 4 Issues with thermal anomaly detection. It is correct, as the authors note in L460, that the issue is "highly controversial", but they do not penetrate the superficial appearance of the widespread controversy. In L461 they use the word "warming". The casual use of this word reveals that they= authors do not understand the physical principle of the radiative nature of the remote sensing signals analyzed by the community.

Answer:

Thanks for your reminding. We misused the inappropriate word "warming" in this sentence. Indeed, the warming phenomenon is just one of the anomalies prior to a main shock. We replaced "warming" with "anomalous" and rewrote the sentence as follows: The anomalous phenomena often occur prior to various earthquake cases, whereas the features of these phenomena are often different.

Q6.

In L512-514 the authors refer to the "unified LAIC model, widely promoted by Sergey Pulinets and his numerous collaborators. However, a close examination of the LAIC

model reveals that it is based on ad hoc assumptions regarding radon. Radon has been proposed to the driver of the LAIC model even though, in the larger context, it is physically impossible that radon can play this role. If radon were responsible to the increase of air ionization prior to majort seismic events, it would have to increase the normal air ion concentration from the "fair weather average" of about 200 per cubic centimeter to 20,000 to 50,000 per ccm. In average crustal rocks, radon is rarer than gold by 6 orders of magnitude. There is about one mole Rn in Earth's atmosphere. Measured close to the ground or in holes in the ground, the pre-earthquake Rn emanation increases by a factor of about 10. Just calculate the number of Rn atoms perccm of normal air and ask yourself, how the decay of these rare Rn atoms can cause a regional increase of the air conductivity by a factor 100-250.

Answer:

Thanks very much for the enlightening quantitative explanation of LAIC model and radon. We rechecked the theories, and modified the description in the paragraph 6 of Section 4 as follows.

Mechanism of pre-seismic thermal anomalies is still inconclusive in the scientific community. Several mechanisms for generation of pre-seismic thermal anomalies detected by satellite have been proposed and aroused a lot of discussion. For example, positive hole theory has been proposed to explain this phenomenon. The electronic charge carriers (positive holes) can be released when the peroxy links break in the stressed rocks, arrive at the Earth's surface and lead to the ionization of air at the ground-air interface. And the recombination of charge carriers at the surface can lead to a spectroscopically distinct, non-thermal IR emission (Freund, 2011; Freund et al., 2009). Besides, a unified LAIC model is proposed, in which the Radon emission in fault zones plays an important role (Molchanov et al., 2004; Pulinets and Ouzounov, 2011). Later, Wu, et al. added the coversphere to the LAIC model after analyzing its importance in the understanding of mechanisms and geophysical processes in earthquake preparation areas (Wu et al., 2016). However, LAIC model is physically impossible based

on the p-hole model. Radon is very rare in the average crustal rocks. Moreover, the measurements of radon emanation on the ground or in the underground shows that Radon emission increases only by a factor of about 10 prior to an earthquake. These insufficient radon atoms cannot bring in a significant increase of the air conductivity by a factor 100-250. Therefore, further validation of these distinctive models is required from physical simulation experiences and synergetic measurements of multiparameter.

References

Chen, S., Liu, P., Guo, Y., Liu, L., & Ma, J. (2015). An experiment on temperature variations in sandstone during biaxial loading. Physics and Chemistry of the Earth, Parts A/B/C, 85-86, 3-8

Freund, F. (2011). Pre-earthquake signals: Underlying physical processes. Journal of Asian Earth Sciences, 41, 383-400

Freund, F.T., Kulahci, I.G., Cyr, G., Ling, J., Winnick, M., Tregloan-Reed, J., & Freund, M.M. (2009). Air ionization at rock surfaces and pre-earthquake signals. Journal of Atmospheric and Solar-Terrestrial Physics, 71, 1824-1834

Molchanov, O., Fedorov, E., Schekotov, A., Gordeev, E., Chebrov, V., Surkov, V., Rozhnoi, A., Andreevsky, S., Iudin, D., Yunga, S., Lutikov, A., Hayakawa, M., & Biagi, P.F. (2004). Lithosphere-atmosphere-ionosphere coupling as governing mechanism for preseismic short-term events in atmosphere and ionosphere. Nat. Hazards Earth Syst. Sci., 4, 757-767

Ouzounov, D., Bryant, N., Logan, T., Pulinets, S., & Taylor, P. (2006). Satellite thermal IR phenomena associated with some of the major earthquakes in 1999–2003. Physics and Chemistry of the Earth, Parts A/B/C, 31, 154-163

Ouzounov, D., Liu, D., Chunli, K., Cervone, G., Kafatos, M., & Taylor, P. (2007). Outgoing long wave radiation variability from IR satellite data prior to major earthquakes. Tectonophysics, 431, 211-220

Pulinets, S., & Ouzounov, D. (2011). Lithosphere–Atmosphere–Ionosphere Coupling (LAIC) model – An unified concept for earthquake precursors validation. Journal of Asian Earth Sciences, 41, 371-382

Pulinets, S.A., Ouzounov, D., Karelin, A.V., Boyarchuk, K.A., & Pokhmelnykh, L.A. (2006). The physical nature of thermal anomalies observed before strong earthquakes. Physics and Chemistry of the Earth, Parts A/B/C, 31, 143-153

Ren, Y., Ma, J., Liu, P., & Chen, S. (2017). Experimental Study of Thermal Field Evolution in the Short-Impending Stage Before Earthquakes. Pure and Applied Geophysics

Tronin, A.A., Hayakawa, M., & Molchanov, O.A. (2002). Thermal IR satellite data application for earthquake research in Japan and China. Journal of Geodynamics, 33, 519-534

Wu, L., Liu, S., Wu, Y., & Wang, C. (2006). Precursors for rock fracturing and failure - Part I: IRR image abnormalities. International Journal of Rock Mechanics and Mining Sciences, 43, 473-482

Please also note the supplement to this comment:
https://www.nat-hazards-earth-syst-sci-discuss.net/nhess-2017-211/nhess-2017-211-AC2-supplement.pdf

———————————————————

---

## Author Response (AR1)

Dear Dr. Malet,

Thank you very much for your comments and suggestions on our manuscript. The manuscript was thoroughly revised according to the problems that you and two reviewers pointed out. We believe that the paper has been improved a lot. Our responses to the comments and suggestions are also enclosed in the PDF file. In the updated manuscript, the modified contents are highlighted with red color.

**Question 1:**

We have now received two extensive reviews and two detailed answers by the authors giving rise to critical enhancements of the quality of the manuscript. Indeed, the main criticism pointed out by the two reviewers is a misunderstanding of the literature by the authors that could be partly linked to the remote-sensing background of the authors. Geophysical knowledge in earth physics and mechanics, radiative transfer and sensing are necessary to be able to critically assess feasibility of detecting and physically understanding infrared thermal anomalies possibly linked to earthquake processes.

…I would suggest the authors to possibly engage discussions with seismologists or earth physicists and possibly integrate them in the authorship of the manuscript. This topic needs a real interdisciplinary discussion.

**Answer:**

Thanks for your help and suggestions. We have discussed with several geophysicists about the mechanism of the pre-seismic anomalies, who have done a lot of work on the pre-seismic anomalies using satellite thermal infrared data and helped to modify the corresponding sections of the manuscript.

**Question 2:**

I have now read twice the original manuscript, and the response of the authors to the referees; it seems that the authors refer to many not scientifically proven theories to discuss the pros and cons of using satellite infrared sensing for crustal ruptures.

**Answer:**

First, these peer-reviewed papers referred in our manuscript are considered as being relatively high quality and scientificalness. The theories in these papers give the possibility to explain the pre-seismic anomaly mechanism, although these explanations may be not perfect.

Second, different scholars support different theories as we described in the paper. These theories can only explain some phenomena of anomalous variations in the earthquake prone regions or are supported by a certain ground or

satellite measurements, but have not been widely validated in the field measurements or are even difficult to verify their rationality in a physically rigorous manner. Thus, the seismic or geophysical communities do not totally accept these theories or mechanisms.

It should be noticed that each physical mechanism has its own physical basis. When these physical mechanisms are used to explain the pre-seismic anomalies, the significant uncertainty becomes an important aspect in the very complex field environment. In particular, it is difficult to verify these theories, which hinders forming a complete logical chain. This is also similar to the mechanism of the earthquake itself that is not very clear, and many different models are used to explain it. The explanations have been added in the updated manuscript.

Finally, the pre-seismic anomaly mechanism is a very big topic that is very complicated and cannot be comprehensively discussed in current paper. Thus, we'd like to discuss such topic deeper in our next coming article.

**Question 3:**

The answers provided are so far implicit and (to my point) vague. It is further difficult to decide on the publication of the manuscript while no revised version has been shared on-line by the authors.

However, I want to help the authors because I am convinced that the topic deserves a state of the art manuscript.

**Answer:**

Thanks for your help. In the E-mail titled "nhess-2017-211 (author) - final response", it says that "To keep manuscript turnover times low we encourage you to submit your responses as soon as possible. Please note that your revised manuscript should not be prepared at this stage. Based on the responses, the Editor will be asked to take a decision about the further handling of your manuscript." Sadly, we misunderstood the email and did not upload the former version of revised manuscript. This time, we earnestly revised the manuscript again. We hope that it could answer the criticisms and satisfy the qualification of NHESS. We believe that our paper could shed some light on this specific study area.

The following are a list of changes marked in red we have made in response to your questions and suggestions.

[revised manuscript text omitted]

---

## Author Response (AR2)

Dear Dr. Malet,

Thank you very much for your comments and suggestions on our manuscript. The manuscript was thoroughly revised according to the problems and suggestions that you and the referee pointed out and offered. We believe that the paper has been improved a lot. Our responses and corresponding corrections are enclosed in the PDF file. In the updated manuscript, the modified contents are highlighted with red color.

**Question 1:**

There are still a few minor corrections to handle which are annotated in the revised version by the referee.

**Answer:**

Thanks for the Prof. Freund's suggestions. We have corrected language mistakes and revised the manuscript according to his suggestions. The whole updated contents can be clearly found in the section of "A list of changes".

**Question 2:**

This paragraph above is incomplete without mentioning the work by Piroddi and Ranieri, which has demonstrated that the thermal anomalies (preceding the L'Aquila earthquake) were clearly associated with the topographic highs (mountain tops) and not with the valley, where all the faults are that could be the source of the so-called greenhouse gases.

**Answer:**

Thanks for the Prof. Freund's suggestions. The revised texts with red color are as follows:

The anomalies in trace gas concentration can be attributed to gas emissions in the lithosphere and photochemical reactions (Cui et al., 2013). The LST anomalies prior to the 2009 Mw 6.3 L'Aquila earthquake illustrate the strong correlation with the topographic height and were clearly distributed over the mountain tops but not the valleys (Piroddi et al., 2014). The anomalous mechanisms of the activation of electronic charge carriers and emission of greenhouse gases or radon gas are considered to demonstrate a consistent manner because of prominent LST anomalies within the limestone regions in this earthquake case. As a result, LST and air temperature may increase, resulting in changes in SLHF and atmospheric water vapor condensation, and propagation to the BT observed by satellite sensors.

**Question 3:**

One important aspect is still missing here: why do aerosols stay suspended in the atmosphere? One very likely

reason is that, due to the generation of predominantly, sometimes exclusively positive airborne ions at the Earth surface, their rise through the atmospheric column and attachment to aerosol particles will kept those particles suspended for longer time. However, since nobody around the world (except Yoshiharu Saito and the PISCO team in Japan) has set up ground stations to monitor the air ionization. That's why there is no independent confirmation that this concept of electrostatic "help" in keeping aerosols suspended.

**Answer:**

According to Prof. Freund's suggestions, the updated texts are as follows:

Therefore, indirect effects of aerosols on the pre-seismic anomalies cannot be ignored. For another thing, the air ionization is a possible reason that keeps aerosols suspended in the atmosphere for many days before the earthquake, which could be explained by the p-hole theory (Freund et al., 2009). The predominantly, sometimes exclusively positive airborne ions generated at the Earth surface rise through the atmospheric column and attach to aerosol particles, thus the ionized aerosols can suspend in the atmosphere for a longer time. More field measurements, which can validate this theory, still need to be carried out.

**Question 4:**

"crustal deformation" cannot cause upwelling. The reason must be different, most certainly thermal. If there is heat input at the ground-to-water interface due to the oxidation of $H_2O$ to $H_2O_2$ by stress-activated p-holes arriving from below (see the Freund et al 2009 paper), we have a physically meaningful mechanism to explain the upwelling.

**Answer:**

The modifications are displayed in the below:

Ouzounov et al. (2006) analyzed the decrease of SST before the Ms 6.8 2003 Boumerdes earthquake and implied that this phenomenon was attributed to the upwelling of cold water from the deep ocean caused by crustal deformation. Freund et al. (2009) disagreed with his explanation, and offered a physically meaningful mechanism to explain the phenomenon of SST anomalies. The p-holes activated by the tectonic stress transfer from the sea floor to the ground-to-water interface and generate the oxidation from $H_2O$ to $H_2O_2$, which changes the thermal radiation balance of the sea surface.

**Question 5:**

I would recommend as well to increase the caption font size of all the figures in particular for the coordinates and names of axes on the maps/graphs.

**Answer:**

Thanks for your suggestions. Except Figure 8, other figures in our paper are referenced from other papers in

order to present the research status. Thus, it is difficult and inappropriate to change the caption font size of the original figures without the authors' approval. We try our best to obtain the high quality figures from the corresponding papers and official publishing websites. However, we still cannot find any higher DPI version for Figure 4 and 9. We'd like to make some font size modification with Photoshop software, if it is OK according to the publishing regulations.

The following contents are a list of changes marked in red we have made in response to your questions and suggestions.

They cause air ionization at the land surface-atmosphere interface when accumulated in a thin surface/subsurface layer, and generate non-thermal infrared emission as a result of the recombination of positive holes. The rock stress adjustment from active faults probably causes anomalies of land surface or air temperatures prior to the earthquake, which could be observed through satellite TIR sensors (Chen et al., 2015; Ren et al., 2017; Wu et al., 2006). The spatiotemporal evolution of the rock temperature field is closely related with its deformation. The rock compression causes

Therefore, much further work need to be carried out to apply this mechanism in the very complex field environment. Other theories also face a similar situation.

We add the following explanation below Table 2:

* NE$\Delta$T denotes the noise equivalent temperature difference.

[revised manuscript text omitted]